# A provable SVD-based algorithm for learning topics in dominant admixture corpus

**Trapit Bansal**†, **C. Bhattacharyya**‡*
Department of Computer Science and Automation
Indian Institute of Science
Bangalore -560012, India
†trapitbansal@gmail.com
‡chiru@csa.iisc.ernet.in

**Ravindran Kannan**
Microsoft Research
India
kannan@microsoft.com

## Abstract

Topic models, such as Latent Dirichlet Allocation (LDA), posit that documents are drawn from admixtures of distributions over words, known as topics. The inference problem of recovering topics from such a collection of documents drawn from admixtures, is NP-hard. Making a strong assumption called *separability*, [4] gave the first provable algorithm for inference. For the widely used LDA model, [6] gave a provable algorithm using clever tensor-methods. But [4, 6] do not learn topic vectors with bounded $l_1$ error (a natural measure for probability vectors).

Our aim is to develop a model which makes intuitive and empirically supported assumptions and to design an algorithm with natural, simple components such as SVD, which provably solves the inference problem for the model with bounded $l_1$ error. A topic in LDA and other models is essentially characterized by a group of co-occurring words. Motivated by this, we introduce topic specific *Catchwords*, a group of words which occur with strictly greater frequency in a topic than any other topic individually and are required to have high frequency together rather than individually. A major contribution of the paper is to show that under this more realistic assumption, which is empirically verified on real corpora, a singular value decomposition (SVD) based algorithm with a crucial pre-processing step of thresholding, can provably recover the topics from a collection of documents drawn from *Dominant admixtures*. Dominant admixtures are convex combination of distributions in which one distribution has a significantly higher contribution than the others. Apart from the simplicity of the algorithm, the sample complexity has near optimal dependence on $w_0$, the lowest probability that a topic is dominant, and is better than [4]. Empirical evidence shows that on several real world corpora, both *Catchwords* and *Dominant admixture* assumptions hold and the proposed algorithm substantially outperforms the state of the art [5].

## 1 Introduction

Topic models [1] assume that each document in a text corpus is generated from an *ad-mixture* of topics, where, each topic is a distribution over words in a Vocabulary. An admixture is a convex combination of distributions. Words in the document are then picked in i.i.d. trials, each trial has a multinomial distribution over words given by the weighted combination of topic distributions. The problem of inference, recovering the topic distributions from such a collection of documents, is provably NP-hard. Existing literature pursues techniques such as variational methods [2] or MCMC procedures [3] for approximating the maximum likelihood estimates.

Given the intractability of the problem one needs further assumptions on topics to derive polynomial time algorithms which can provably recover topics. A possible (strong) assumption is that each document has only one topic but the collection can have many topics. A document with only one topic is sometimes referred as a *pure topic* document. [7] proved that a natural algorithm, based on SVD, recovers topics when each document is pure and in addition, for each topic, there is a set of words, called *primary words*, whose total frequency in that topic is close to 1. More recently, [6] show using tensor methods that if the topic weights have Dirichlet distribution, we can learn the topic matrix. Note that while this allows non-pure documents, the Dirichlet distribution gives essentially uncorrelated topic weights.

In an interesting recent development [4, 5] gave the first provable algorithm which can recover topics from a corpus of documents drawn from admixtures, assuming *separability*. Topics are said to be separable if in every topic there exists at least one *Anchor* word. A word in a topic is said to be an *Anchor* word for that topic if it has a high probability in that topic and *zero* probability in remaining topics. The requirement of high probability in a topic for a single word is unrealistic.

**Our Contributions:** Topic distributions, such as those learnt in LDA, try to model the co-occurrence of a group of words which describes a theme. Keeping this in mind we introduce the notion of *Catchwords*. A group of words are called *Catchwords* of a topic, if each word occurs strictly more frequently in the topic than other topics and together they have high frequency. This is a much weaker assumption than separability. Furthermore we observe, empirically, that posterior topic weights assigned by LDA to a document often have the property that one of the weights is significantly higher than the rest. Motivated by this observation, which has not been exploited by topic modeling literature, we suggest a new assumption. It is natural to assume that in a text corpus, a document, even if it has multiple themes, will have an overarching dominant theme. In this paper we focus on document collections drawn from *dominant admixtures*. A document collection is said to be drawn from a dominant admixture if for every document, there is one topic whose weight is significantly higher than the other topics and in addition, for every topic, there is a small fraction of documents which are nearly purely on that topic. The main contribution of the paper is to show that under these assumptions, our algorithm, which we call TSVD, indeed provably finds a good approximation in total $l_1$ error to the topic matrix. We prove a bound on the error of our approximation which does not grow with dictionary size $d$, unlike [5] where the error grows linearly with $d$.

Empirical evidence shows that on semi-synthetic corpora constructed from several real world datasets, as suggested by [5], TSVD substantially outperforms the state of the art [5]. In particular it is seen that compared to [5] TSVD gives 27% lower error in terms of $l_1$ recovery on 90% of the topics.

**Problem Definition:** $d, k, s$ will denote respectively, the number of words in the dictionary, number of topics and number of documents. $d, s$ are large, whereas, $k$ is to be thought of as much smaller. Let $\mathcal{S}_k = \{x = (x_1, x_2, \ldots, x_k) : x_l \geq 0; \sum_l x_l = 1\}$. For each topic, there is a fixed vector in $\mathcal{S}_k$ giving the probability of each word in that topic. Let $\mathbf{M}$ be the $d \times k$ matrix with these vectors as its columns.

Documents are picked in i.i.d. trials. To pick document $j$, one first picks a $k$-vector $W_{1j}, W_{2j}, \ldots, W_{kj}$ of topic weights according to a fixed distribution on $\mathcal{S}_k$. Let $P_{.,j} = \mathbf{M}W_{.,j}$ be the weighted combination of the topic vectors. Then the $m$ words of the document are picked in i.i.d. trials; each trial picks a word according to the multinomial distribution with $P_{.,j}$ as the probabilities. All that is given as data is the frequency of words in each document, namely, we are given the $d \times s$ matrix $\mathbf{A}$, where $A_{ij} = \frac{\text{Number of occurrences of word } i \text{ in Document } j}{m}$. Note that $E(\mathbf{A}|\mathbf{W}) = \mathbf{P}$, where, the expectation is taken entry-wise.

*In this paper we consider the problem of finding* $\mathbf{M}$ *given* $\mathbf{A}$.

## 2 Previous Results

In this section we review literature related to designing provable algorithms for topic models. For an overview of topic models we refer the reader to the excellent survey of [1]. Provable algorithms for recovering topic models was started by [7]. Latent Semantic Indexing (LSI) [8] remains a successful method for retrieving similar documents by using SVD. [7] showed that one can recover $\mathbf{M}$ from a

collection of documents, with *pure topics*, by using SVD based procedure under the additional Primary Words assumption. [6] showed that in the admixture case, if one assumes Dirichlet distribution for the topic weights, then, indeed, using tensor methods, one can learn $\mathbf{M}$ to $l_2$ error provided some added assumptions on numerical parameters like condition number are satisfied.

The first provably polynomial time algorithm for admixture corpus was given in [4, 5]. For a topic $l$, a word $i$ is an **anchor word** if: $M_{i,l} \geq p_0$ and $M_{i,l'} = 0 \quad \forall l' \neq l$.

**Theorem 2.1** *[4] If every topic has an anchor word, there is a polynomial time algorithm that returns an $\hat{M}$ such that with high probability,*

$$\sum_{l=1}^{k} \sum_{i=1}^{d} |\hat{M}_{il} - M_{il}| \leq d\varepsilon \quad provided \ \ s \geq Max \left\{ O\left( \frac{k^6 \log d}{a^4 \varepsilon^2 p_0^6 \gamma^2 m} \right), O\left( \frac{k^4}{\gamma^2 a^2} \right) \right\},$$

*where, $\gamma$ is the condition number of $E(WW^T)$, $a$ is the minimum expected weight of a topic and $m$ is the number of words in each document.*

Note that the error grows linearly in the dictionary size $d$, which is often large. Note also the dependence of $s$ on parameters $p_0$, which is, $1/p_0^6$ and on $a$, which is $1/a^4$. If, say, the word "run" is an anchor word for the topic "baseball" and $p_0 = 0.1$, then the requirement is that every 10 th word in a document on this topic is "run". This seems too strong to be realistic. It would be more realistic to ask that a set of words like - "run", "hit", "score", etc. together have frequency at least 0.1 which is what our catchwords assumption does.

# 3   Learning Topics from Dominant Admixtures

Informally, a document is said to be drawn from a Dominant Admixture if the document has one *dominant topic*. Besides its simplicity, we show empirical evidence from real corpora to demonstrate that topic dominance is a reasonable assumption. The Dominant Topic assumption is weaker than the Pure Topic assumption. More importantly, SVD based procedures proposed by [7] will not apply. Inspired by the *Primary Words* assumption we introduce the assumption that each topic has a set of *Catchwords* which individually occur more frequently in that topic than others. This is again a much weaker assumption than both *Primary Words* and *Anchor Words* assumptions and can be verified experimentally. In this section we establish that by applying SVD on a matrix, obtained by thresholding the word-document matrix, and subsequent $k$-means clustering can learn topics having Catchwords from a Dominant Admixture corpus.

## 3.1   Assumptions: Catchwords and Dominant admixtures

Let $\alpha, \beta, \rho, \delta, \varepsilon_0$ be non-negative reals satisfying: $\beta + \rho \leq (1 - \delta)\alpha, \quad \alpha + 2\delta \leq 0.5, \quad \delta \leq 0.08$

**Dominant topic Assumption** (a) For $j = 1, 2, \ldots, s$, document $j$ has a dominant topic $l(j)$ such that $W_{l(j),j} \geq \alpha$ and $W_{l'j} \leq \beta, \quad \forall l' \neq l(j)$.

(b) For each topic $l$, there are at least $\varepsilon_0 w_0 s$ documents in each of which topic $l$ has weight at least $1 - \delta$.

**Catchwords Assumption:** There are $k$ disjoint sets of words - $S_1, S_2, \ldots, S_k$ such that with $\varepsilon$ defined in (5), $\forall i \in S_l, \forall l' \neq l, M_{il'} \leq \rho M_{il}, \quad \sum_{i \in S_l} M_{il} \geq p_0$,

$$\forall i \in S_l, \ m\delta^2 \alpha M_{il} \geq 8 \ln \left( \frac{20}{\varepsilon w_0} \right). \tag{1}$$

Part (b) of the Dominant Topic Assumption is in a sense necessary for "identifiability" - namely for the model to have a set of $k$ document vectors so that every document vector is in the convex hull of these vectors. The Catchwords assumption is natural to describe a theme as it tries to model a unique group of words which is likely to co-occur when a theme is expressed. This assumption is close to topics discovered by LDA like models, which try to model co-occurence of words. If $\alpha, \delta \in \Omega(1)$, then, the assumption (1) says $M_{il} \in \Omega^*(1/m)$. In fact if $M_{il} \in o(1/m)$, we do not expect to see word $i$ (in topic $l$), so it cannot be called a catchword at all.

A slightly different (but equivalent) description of the model will be useful to keep in mind. What is fixed (not stochastic) are the matrices $\mathbf{M}$ and the distribution of the weight matrix $\mathbf{W}$. To pick document $j$, we can first pick the dominant topic $l$ in document $j$ and condition the distribution of $W_{\cdot,j}$ on this being the dominant topic. One could instead also think of $W_{\cdot,j}$ being picked from a mixture of $k$ distributions. Then, we let $P_{ij} = \sum_{l=1}^{k} M_{il} W_{lj}$ and pick the $m$ words of the document in i.i.d multinomial trials as before. We will assume that

$$T_l = \{j : l \text{ is the dominant topic in document } j\} \text{ satisfies } |T_l| = w_l s,$$

where, $w_l$ is the probability of topic $l$ being dominant. This is only approximately valid, but the error is small enough that we can disregard it.

For $\zeta \in \{0, 1, 2, \ldots, m\}$, let $p_i(\zeta, l)$ be the probability that $j \in T_l$ and $A_{ij} = \zeta/m$ and $q_i(\zeta, l)$ the corresponding "empirical probability":

$$p_i(\zeta, l) = \int_{W_{\cdot,j}} \binom{m}{\zeta} P_{ij}^{\zeta}(1 - P_{ij})^{m-\zeta} \text{Prob}(W_{\cdot,j} \mid j \in T_l) \text{ Prob}(j \in T_l), \text{ where, } P_{\cdot,j} = \mathbf{M} W_{\cdot,j}. \tag{2}$$

$$q_i(\zeta, l) = \frac{1}{s} |\{j \in T_l : A_{ij} = \zeta/m\}|. \tag{3}$$

Note that $p_i(\zeta, l)$ is a real number, whereas, $q_i(\zeta, l)$ is a random variable with $E(q_i(\zeta, l)) = p_i(\zeta, l)$. We need a technical assumption on the $p_i(\zeta, l)$ (which is weaker than unimodality).

**No-Local-Min Assumption:** We assume that $p_i(\zeta, l)$ does not have a local minimum, in the sense:

$$p_i(\zeta, l) \; > \; \text{Min}(p_i(\zeta - 1, l), p_i(\zeta + 1, l)) \; \forall \; \zeta \in \{1, 2, \ldots, m - 1\}. \tag{4}$$

The justification for this assumption is two-fold. First, generally, Zipf's law kind of behaviour where the number of words plotted against relative frequencies declines as a power function has often been observed. Such a plot is monotonically decreasing and indeed satisfies our assumption. But for Catchwords, we do not expect this behaviour - indeed, we expect the curve to go up initially as the relative frequency increases, then reach a maximum and then decline. This is a unimodal function and also satisfies our assumption.

**Relative sizes of parameters:** Before we close this section, a discussion on the values of the parameters is in order. Here, $s$ is large. For asymptotic analysis, we can think of it as going to infinity. $1/w_0$ is also large and can be thought of as going to infinity. [In fact, if $1/w_0 \in O(1)$, then, intuitively, we see that there is no use of a corpus of more than constant size - since our model has i.i.d. documents, intuitively, the number of samples we need should depend mainly on $1/w_0$]. $m$ is (much) smaller, but need not be constant.

$c$ refers to a generic constant independent of $m, s, 1/w_0, \varepsilon, \delta$; its value may be different in different contexts.

## 3.2 The TSVD Algorithm

Existing SVD based procedures for clustering on raw word-document matrices fail because the spread of frequencies of a word within a topic is often more (at least not significantly less) than the gap between the word's frequencies in two different topics. Hypothetically, the frequency for the word *run*, in the topic *Sports*, may range upwards of 0.01, say. But in other topics, it may range from, say, 0 to 0.005. The success of the algorithm will lie on correctly identifying the dominant topics such as sports by identifying that the word *run* has occurred with high frequency. In this example, the gap (0.01-0.005) between Sports and other topics is less than the spread within Sports (1.0-0.01), so a 2-clustering approach (based on SVD) will split the topic Sports into two. While this is a toy example, note that if we threshold the frequencies at say 0.01, ideally, sports will be all above and the rest all below the threshold, making the succeeding job of clustering easy.

There are several issues in extending beyond the toy case. Data is not one-dimensional. We will use different thresholds for each word; word $i$ will have a threshold $\zeta_i/m$. Also, we have to compute $\zeta_i/m$. Ideally we would not like to split any $T_l$, namely, we would like that for each $l$ and and each $i$, either most $j \in T_l$ have $A_{ij} > \zeta_i/m$ or most $j \in T_l$ have $A_{ij} \leq \zeta_i/m$. We will show that

our threshold procedure indeed achieves this. One other nuance: to avoid conditioning, we split the data $\mathbf{A}$ into two parts $\mathbf{A}^{(1)}$ and $\mathbf{A}^{(2)}$, compute the thresholds using $\mathbf{A}^{(1)}$ and actually do the thresholding on $\mathbf{A}^{(2)}$. We will assume that the intial $\mathbf{A}$ had $2s$ columns, so each part now has $s$ columns. Also, $T_1, T_2, \ldots, T_k$ partitions the columns of $\mathbf{A}^{(1)}$ as well as those of $\mathbf{A}^{(2)}$. The columns of thresholded matrix $\mathbf{B}$ are then clustered, by a technique we call Project and Cluster, namely, we project the columns of $\mathbf{B}$ to its $k-$dimensional SVD subspace and cluster in the projection. The projection before clustering has recently been proven [9] (see also [10]) to yield good starting cluster centers. The clustering so found is not yet satisfactory. We use the classic Lloyd's $k$-means algorithm proposed by [12]. As we will show, the partition produced after clustering, $\{R_1, \ldots, R_k\}$ of $\mathbf{A}^{(2)}$ is close to the partition induced by the Dominant Topics, $\{T_1, \ldots, T_k\}$. Catchwords of topic $l$ are now (approximately) identified as the most frequently occurring words in documents in $R_l$. Finally, we identify nearly pure documents in $T_l$ (approximately) as the documents in which the catchwords occur the most. Then we get an approximation to $M_{.,l}$ by averaging these nearly pure documents. We now describe the precise algorithm.

### 3.3  Topic recovery using Thresholded SVD

**Threshold SVD based K-means (TSVD)**

$$\varepsilon = \text{Min}\ \left( \frac{1}{900c_0^2} \frac{\alpha p_0}{k^3 m}\ ,\ \frac{\varepsilon_0 \sqrt{\alpha p_0}\delta}{640 m \sqrt{k}}\ ,\ \right). \tag{5}$$

1. Randomly partition the columns of $\mathbf{A}$ into two matrices $\mathbf{A}^{(1)}$ and $\mathbf{A}^{(2)}$ of $s$ columns each.

2. **Thresholding**

   (a) **Compute Thresholds on $\mathbf{A}^{(1)}$** For each $i$, let $\zeta_i$ be the highest value of $\zeta \in \{0, 1, 2, \ldots, m\}$ such that $|\{j : A_{ij}^{(1)} > \frac{\zeta}{m}\}| \geq \frac{w_0}{2} s$; $|\{j : A_{ij}^{(1)} = \frac{\zeta}{m}\}| \leq 3\varepsilon w_0 s$.

   (b) **Do the thresholding on $\mathbf{A}^{(2)}$:** $B_{ij} = \begin{cases} \sqrt{\zeta_i} & \text{if } A_{ij}^{(2)} > \zeta_i/m \text{ and } \zeta_i \geq 8\ln(20/\varepsilon w_0) \\ 0 & \text{otherwise} \end{cases}$.

3. **SVD** Find the best rank $k$ approximation $\mathbf{B}^{(k)}$ to $\mathbf{B}$.

4. **Identify Dominant Topics**

   (a) **Project and Cluster** Find (approximately) optimal $k$-means clustering of the columns of $\mathbf{B}^{(k)}$.

   (b) **Lloyd's Algorithm** Using the clustering found in Step 4(a) as the starting clustering, apply Lloyd's $k$-means algorithm to the columns of $\mathbf{B}$ ($\mathbf{B}$, not $\mathbf{B}^{(k)}$).

   (c) Let $R_1, R_2, \ldots, R_k$ be the $k-$partition of $[s]$ corresponding to the clustering after Lloyd's. *//*Will prove that $R_l \approx T_l$*//*

5. **Identify Catchwords**

   (a) For each $i, l$, compute $g(i, l) =$ the $(\lfloor \varepsilon_0 w_0 s/2 \rfloor)$th highest element of $\{A_{ij}^{(2)} : j \in R_l\}$.

   (b) Let $J_l = \left\{ i : g(i, l) > \text{Max}\left( \frac{4}{m\delta^2} \ln(20/\varepsilon w_0), \text{Max}_{l' \neq l} \gamma\, g(i, l') \right) \right\}$, where, $\gamma = \frac{1-2\delta}{(1+\delta)(\beta+\rho)}$.

6. **Find Topic Vectors** Find the $\lfloor \varepsilon_0 w_0 s/2 \rfloor$ highest $\sum_{i \in J_l} A_{ij}^{(2)}$ among all $j \in [s]$ and return the average of these $A_{.,j}$ as our approximation $\hat{M}_{.,l}$ to $M_{.,l}$.

**Theorem 3.1  Main Theorem** *Under the Dominant Topic, Catchwords and No-Local-Min assumptions, the algorithm succeeds with high probability in finding an $\hat{M}$ so that*

$$\sum_{i,l} |M_{il} - \hat{M}_{il}| \in O(k\delta),\ \textit{provided }^{[1]} s \in \Omega^* \left( \frac{1}{w_0} \left( \frac{k^6 m^2}{\alpha^2 p_0^2} + \frac{m^2 k}{\varepsilon_0^2 \delta^2 \alpha p_0} + \frac{d}{\varepsilon_0 \delta^2} \right) \right).$$

A note on the sample complexity is in order. Notably, the dependence of $s$ on $w_0$ is best possible (namely $s \in \Omega^*(1/w_0)$) within logarithmic factors, since, if we had fewer than $1/w_0$ documents, a topic which is dominant with probability only $w_0$ may have none of the documents in the collection. The dependence of $s$ on $d$ needs to be at least $d/\varepsilon_0 w_0 \delta^2$: to see this, note that we only assume that there are $r = O(\varepsilon_0 w_0 s)$ nearly pure documents on each topic. Assuming we can find this set (the algorithm approximately does), their average has standard deviation of about $\sqrt{M_{il}}/\sqrt{r}$ in coordinate $i$. If topic vector $M_{.,l}$ has $O(d)$ entries, each of size $O(1/d)$, to get an approximation of $M_{.,l}$ to $l_1$ error $\delta$, we need the per coordinate error $1/\sqrt{dr}$ to be at most $\delta/d$ which implies $s \geq d/\varepsilon_0 w_0 \delta^2$. Note that to get comparable error in [4], we need a quadratic dependence on $d$.

There is a long sequence of Lemmas to prove the theorem. To improve the readability of the paper we relegate the proofs to supplementary material [14]. The essence of the proof lies in proving that the clustering step correctly identifies the partition induced by the dominant topics. For this, we take advantage of a recent development on the $k-$means algorithm from [9] [see also [10]], where, it is shown that under a condition called the *Proximity Condition*, Lloyd's $k$ means algorithm starting with the centers provided by the SVD-based algorithm, correctly identifies almost all the documents' dominant topics. We prove that indeed the Proximity Condition holds. This calls for machinery from Random Matrix theory (in particular bounds on singular values). We prove that the singular values of the thresholded word-document matrix are nicely bounded. Once the dominant topic of each document is identified, we are able to find the Catchwords for each topic. Now, we rely upon part (b.) of the Dominant Topic assumption : that is there is a small fraction of nearly Pure Topic-documents for each topic. The Catchwords help isolate the nearly pure-topic documents and hence find the topic vectors. The proofs are complicated by the fact that each step of the algorithm induces conditioning on the data – for example, after clustering, the document vectors in one cluster are not independent anymore.

## 4 Experimental Results

We compare the thresholded SVD based k-means (**TSVD**[2]) algorithm 3.3 with the algorithms of [5], *Recover-KL* and *Recover-L2*, using the code made available by the authors[3]. We observed the results of Recover-KL to be better than Recover-L2, and report here the results of Recover-KL (abbreviated R-KL), full set of results can be found in supplementary section 5. We first provide empirical support for the algorithm assumptions in Section 3.1, namely the dominant topic and the catchwords assumption. Then we show on 4 different semi-synthetic data that TSVD provides as good or better recovery of topics than the Recover algorithms. Finally on real-life datasets, we show that the algorithm performs as well as [5] in terms of perplexity and topic coherence.

**Implementation Details:** TSVD parameters ($w_0,\ \varepsilon,\ \varepsilon_0,\ \gamma$) are not known in advance for real corpus. We tested empirically for multiple settings and the following values gave the best performance. Thresholding parameters used were: $w_0 = \frac{1}{k}, \varepsilon = \frac{1}{6}$. For finding the catchwords, $\gamma = 1.1, \varepsilon_0 = \frac{1}{3}$ in step 5. For finding the topic vectors (step 6), taking the top 50% ($\varepsilon_0 w_0 = \frac{1}{k}$) gave empirically better results. The same values were used on all the datasets tested. The new algorithm is sensitive to the initialization of the first k-means step in the projected SVD space. To remedy this, we run 10 independent random initializations of the algorithm with K-Means++ [13] and report the best result.

**Datasets:** We use four real word datasets in the experiments. As pre-processing steps we removed standard stop-words, selected the vocabulary size by term-frequency and removed documents with less than 20 words. Datasets used are: (1) **NIPS**[4]: Consists of 1,500 NIPS full papers, vocabulary of 2,000 words and mean document length 1023. (2) **NYT**[4]: Consists of a random subset of 30,000 documents from the New York Times dataset, vocabulary of 5,000 words and mean document length 238. (3) **Pubmed**[4]: Consists of a random subset of 30,000 documents from the Pubmed abstracts dataset, vocabulary of 5,030 words and mean document length 58. (4) **20NewsGroup**[5] (20NG): Consist of 13,389 documents, vocabulary of 7,118 words and mean document length 160.

| Corpus | $s$ | $k$ | % s with Dominant Topics ($\alpha = 0.4$) | % s with Pure Topics ($\delta = 0.05$) | % Topics with CW | CW Mean Frequency |
|---|---|---|---|---|---|---|
| NIPS | 1500 | 50 | 56.6% | 2.3% | 96% | 0.05 |
| NYT | 30000 | 50 | 63.7% | 8.5% | 98% | 0.07 |
| Pubmed | 30000 | 50 | 62.2% | 5.1% | 78% | 0.05 |
| 20NG | 13389 | 20 | 74.1% | 39.5% | 85% | 0.06 |

Table 1: Algorithm Assumptions. For dominant topic assumption, fraction of documents with satisfy the assumption for $(\alpha, \beta) = (0.4, 0.3)$ are shown. % documents with almost pure topics ($\delta = 0.05$, i.e. 95% pure) are also shown. Last two columns show results for catchwords (CW) assumption.

## 4.1 Algorithm Assumptions

To check the *dominant topic* and *catchwords* assumptions, we first run 1000 iterations of Gibbs sampling on the real corpus and learn the posterior document-topic distribution ($\{W_{\cdot,j}\}$) for each document in the corpus (by averaging over 10 saved-states separated by 50 iterations after the 500 burn-in iterations). We will use this posterior document-topic distribution as the document generating distribution to check the two assumptions.

**Dominant topic assumption:** Table 1 shows the fraction of the documents in each corpus which satisfy this assumption with $\alpha = 0.4$ (minimum probability of dominant topic) and $\beta = 0.3$ (maximum probability of non-dominant topics). The fraction of documents which have almost pure topics with highest topic weight at least 0.95 ($\delta = 0.05$) is also shown. The results indicate that the dominant topic assumption is well justified (on average 64% documents satisfy the assumption) and there is also a substantial fraction of documents satisfying almost pure topic assumption.

**Catchwords assumption:** We first find a $k$-clustering of the documents $\{T_1, \dots, T_k\}$ by assigning all documents which have highest posterior probability for the same topic into one cluster. Then we use step 5 of TSVD (Algorithm 3.3) to find the set of catchwords for each topic-cluster, i.e. $\{S_1, \dots, S_k\}$, with the parameters: $\epsilon_0 w_0 = \frac{1}{3k}$, $\gamma = 2.3$ (taking into account constraints in Section 3.1, $\alpha = 0.4, \beta = 0.3, \delta = 0.05, \rho = 0.07$). Table 1 reports the fraction of topics with non-empty set of catchwords and the average per topic frequency of the catchwords[6]. Results indicate that most topics on real data contain catchwords (Table 1, second-last column). Moreover, the average per-topic frequency of the group of catchwords for that topic is also quite high (Table 1, last column).

## 4.2 Empirical Results

**Semi-synthetic Data:** Following [5], we generate semi-synthetic corpora from LDA model trained by MCMC, to ensure that the synthetic corpora retain the characteristics of real data. Gibbs sampling[7] is run for 1000 iterations on all the four datasets and the final word-topic distribution is used to generate varying number of synthetic documents with document-topic distribution drawn from a symmetric Dirichlet with hyper-parameter 0.01. For NIPS, NYT and Pubmed we use $k = 50$ topics, for 20NewsGroup $k = 20$, and mean document lengths of 1000, 300, 100 and 200 respectively. Note that the synthetic data is *not* guaranteed to satisfy dominant topic assumption for every document (on average about 80% documents satisfy the assumption for value of $(\alpha, \beta)$ tested in Section 4.1).

**Topic Recovery on Semi-synthetic Data:** We learn the word-topic distribution ($\hat{M}$) for the semi-synthetic corpora using TSVD and the Recover algorithms of [5]. Given these learned topic distributions and the original data-generating distribution ($M$), we align the topics of $M$ and $\hat{M}$ by bipartite matching and evaluate the $l_1$ distance between each pair of topics. We report the average of $l_1$ error across topics (called $l_1$ *reconstruction-error* [5]) in Table 2 for TSVD and Recover-KL (R-KL). TSVD has smaller error on most datasets than the R-KL algorithm. We observed performance of TSVD to be always better than Recover-L2 (see supplement Table 1 for full results). Best performance is observed on NIPS which has largest mean document length, indicating that larger $m$ leads to better recovery. Results on 20NG are slightly worse than R-KL for smaller sample size, but performance improves for larger number of documents. While the error-values in Table 2 are

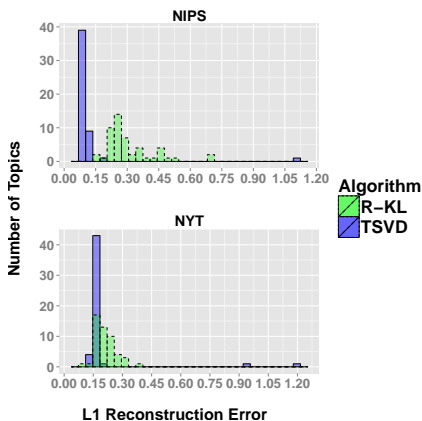

| Corpus | Documents | R-KL | TSVD |
|--------|-----------|------|------|
| **NIPS** | 40,000 | 0.308 | **0.115** (62.7%) |
| | 50,000 | 0.308 | **0.145** (52.9%) |
| | 60,000 | 0.311 | **0.131** (57.9%) |
| **Pubmed** | 40,000 | 0.332 | **0.288** (13.3%) |
| | 50,000 | 0.326 | **0.280** (14.1%) |
| | 60,000 | 0.328 | **0.284** (13.4%) |
| **20NG** | 40,000 | **0.120** | 0.124 (-3.3%) |
| | 50,000 | 0.114 | **0.113** (0.9%) |
| | 60,000 | 0.110 | **0.106** (3.6%) |
| **NYT** | 40,000 | 0.208 | **0.195** (6.3%) |
| | 50,000 | 0.206 | **0.185** (10.2%) |
| | 60,000 | 0.200 | **0.194** (3.0%) |

Figure 1: Histogram of $l_1$ error across topics (40,000 documents). *TSVD*(blue, solid border) gets smaller error on most topics than *R-KL*(green, dashed border).

Table 2: $l_1$ reconstruction error on various semi-synthetic datasets. Brackets in the last column give percent improvement over R-KL (best performing Recover algorithm). Full results in supplementary.

averaged values across topics, Figure 1 shows that TSVD algorithm achieves much better topic recovery for majority of the topics (>90%) on most datasets (overall average improvement of 27%, full results in supplement Figure 1).

**Topic Recovery on Real Data:** To evaluate perplexity [2] on real data, the held-out sets consist of 350 documents for NIPS, 10000 documents for NYT and Pubmed, and 6780 documents for 20NewsGroup. *TSVD* achieved perplexity measure of **835 (NIPS), 1307 (Pubmed), 1555 (NYT), 2390 (20NG)** while *Recover-KL* achieved **754 (NIPS), 1188 (Pubmed), 1579 (NYT), 2431 (20NG)** (refer to supplement Table 2 for complete results). TSVD gives comparable perplexity with Recover-KL, results being slightly better on NYT and 20NewsGroup which are larger datasets with moderately high mean document lengths. We also find comparable results on Topic Coherence [11] (see Table 2 in supplementary for topic coherence results and Table 3 for list of top words of topics).

**Summary:** We evaluated the proposed algorithm, **TSVD**, rigorously on multiple datasets with respect to the state of the art [5] (Recover-KL and Recover-L2), following the evaluation methodology of [5]. In Table 2 we show that the $l_1$ reconstruction error for the new algorithm is small and on average **19.6%** better than the best results of the Recover algorithms [5]. In Figure 1, we show that TSVD achieves significantly better recover on majority of the topics. We also demonstrate that on real datasets the algorithm achieves comparable perplexity and topic coherence to Recover algorithms. Moreover, we show on multiple real world datasets that the algorithm assumptions are well justified in practice.

## Conclusion

Real world corpora often exhibits the property that in every document there is one topic dominantly present. A standard SVD based procedure will not be able to detect these topics, however TSVD, a thresholded SVD based procedure, as suggested in this paper, discovers these topics. While SVD is time-consuming, there have been a host of recent sampling-based approaches which make SVD easier to apply to massive corpora which may be distributed among many servers. We believe that apart from topic recovery, thresholded SVD can be applied even more broadly to similar problems, such as matrix factorization, and will be the basis for future research.

**Acknowledgements** TB was supported by a Department of Science and Technology (DST) grant.

## Footnotes

*http://mllab.csa.iisc.ernet.in/tsvd

[1]The superscript $^*$ hides a logarithmic factor in $dsk/\delta_{\text{fail}}$, where, $\delta_{\text{fail}} > 0$ is the desired upper bound on the probability of failure.

[2] Resources available at: http://mllab.csa.iisc.ernet.in/tsvd

[3] http://www.cs.nyu.edu/~halpern/files/anchor-word-recovery.zip

[4] http://archive.ics.uci.edu/ml/datasets/Bag+of+Words

[5] http://qwone.com/~jason/20Newsgroups

[6] $\left( \frac{1}{k} \sum_{l=1}^{k} \frac{1}{|T_l|} \sum_{i \in S_l} \sum_{j \in T_l} A_{ij} \right)$

[7] Dirichlet hyperparameters used: document-topic = 0.03 and topic-word = 1

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
