[Supplementary Material · supplementary.pdf]

# Supplementary for "A provable SVD-based algorithm for learning topics in dominant admixture corpus"

**Trapit Bansal†, C. Bhattacharyya‡**
Department of Computer Science and Automation
Indian Institute of Science
Bangalore -560012, India
†trapitbansal@gmail.com
‡chiru@csa.iisc.ernet.in

**Ravindran Kannan**
Microsoft Research
India
kannan@microsoft.com

## 1  Introduction

In this supplement we recall the TSVD algorithm, neccessary definitions, and prove the correctness of TSVD. We also present additional experimental results to support the main claims in the original paper

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

## 3   Line of Proof

We describe the Lemmas we prove to establish the result. The detailed proofs are in the Section 4.

### 3.1   General Facts

We start with a consequence of the no-local-minimum assumption. We use that assumption solely through this Lemma.

**Lemma 3.1** *Let $p_i(\zeta, l)$ be as in (6). If for some $\zeta_0 \in \{0, 1, \ldots, m\}$ and $\nu \geq 0$, $\sum_{\zeta \geq \zeta_0} p_i(\zeta, l) \geq \nu$ and also $\sum_{\zeta \leq \zeta_0} p_i(\zeta, l) \geq \nu$ then, $p_i(\zeta_0, l) \geq \frac{\nu}{m}$.*

Next, we state a technical Lemma which is used repeatedly. It states that for every $i, \zeta, l$, the empirical probability that $A_{ij} = \zeta/m$ is close to the true probability. Unsurprisingly, we prove it using H-C. But we will state a consequence in the form we need in the sequel.

**Lemma 3.2** Let $p_i(\zeta, l)$ and $q_i(\zeta, l)$ be as in (6) and (7). We have

$$\forall i, l, \zeta : Prob\left(|p_i(\zeta, l) - q_i(\zeta, l)| \geq \frac{\varepsilon}{2}\sqrt{w_0}\sqrt{p_i(\zeta, l)} + \frac{\varepsilon^2 w_0}{2}\right) \quad \leq \ 2\exp(-\varepsilon^2 s w_0/8).$$

*From this it follows that with probability at least* $1 - 2\exp(-\varepsilon^2 w_0 s/8)$,

$$\frac{1}{2}q_i(\zeta, l) - \varepsilon^2 w_0 \quad \leq \ p_i(\zeta, l) \quad \leq \ 2q_i(\zeta, l) + 2\varepsilon^2 w_0.$$

### 3.1.1 Properties of Thresholding

Say that a threshold $\zeta_i$ "splits" $T_l^{(2)}$ if $T_l^{(2)}$ has a significant number of $j$ with $A_{ij} > \zeta_i/m$ and also a significant number of $j$ with $A_{ij} \leq \zeta_i/m$. Intuitively, it would be desirable if no threshold splits any $T_l$, so that, in $\mathbf{B}$, for each $i, l$, either most $j \in T_l^{(2)}$ have $B_{ij} = 0$ or most $j \in T_l^{(2)}$ have $B_{ij} = \sqrt{\zeta_i}$. We now prove that this is indeed the case with proper bounds. We henceforth refer to the conclusion of the Lemma below by the mnemonic "no threshold splits any $T_l$".

**Lemma 3.3** (**No Threshold Splits any** $T_l$) *For a fixed* $i, l$, *with probability at least* $1 - 2\exp(-\varepsilon^2 w_0 s/8)$, *the following holds:*

$$Min\ \left(Prob(A_{ij}^{(2)} \leq \frac{\zeta_i}{m} \ ; j \in T_l^{(2)}), \quad Prob(A_{ij}^{(2)} > \frac{\zeta_i}{m} \ ; j \in T_l^{(2)})\right) \quad \leq 4m\varepsilon w_0.$$

Let $\mu$ be a $d \times s$ matrix whose columns are given by

$$\forall j \in T_l^{(2)} \ , \ \mu_{\cdot,j} = E(B_{\cdot,j} \mid j \in T_l).$$

$\mu$ 's columns corresponding to all $j \in T_l$ are the same. The entries of the matrix $\mu$ are fixed (real numbers) once we have $\mathbf{A}^{(1)}$ (and the thresholds $\zeta_i$ are determined). Note: We have "integrated out $W$", i.e.,

$$\mu_{ij} = \int_{W_{\cdot,j}} \text{Prob}(W_{\cdot,j} | j \in T_l) E(B_{ij} | W_{\cdot,j}).$$

(So, think of $W_{\cdot,j}$ for $\mathbf{A}^{(1)}$ 's columns being picked first from which $\zeta_i$ is calculated. $W_{\cdot,j}$ for columns of $\mathbf{A}^{(2)}$ are not yet picked until the $\zeta_i$ are determined.) But $\mu_{ij}$ are random variables before we fix $\mathbf{A}^{(1)}$. The following Lemma is a direct consequence of "no threshold splits any $T_l$".

**Lemma 3.4** Let $\zeta_i' = Max(\zeta_i, 8\ln(20/\varepsilon w_0))$. With probability at least $1 - 4kd\exp(-\varepsilon^2 s w_0/8)$ (over the choice of $\mathbf{A}^{(1)}$):

$$\forall l, \forall j \in T_l, \forall i : \mu_{ij} \leq \varepsilon_l\sqrt{\zeta_i'} \ OR \ \mu_{ij} \geq \sqrt{\zeta_i'}(1 - \varepsilon_l)$$
$$\forall l, \forall i, Var(B_{ij}) \leq 2\varepsilon_l \zeta_i', \tag{10}$$

where, $\varepsilon_l = 4m\varepsilon w_0/w_l$.

So far, we have proved that for every $i$, the threshold does not split any $T_l$. But this is not sufficient in itself to be able to cluster (and hence identify the $T_l$), since, for example, this alone does not rule out the extreme cases that for most $j$ in every $T_l$, $A_{ij}^{(2)}$ is above the threshold (whence $\mu_{ij} \geq (1-\varepsilon_l)\sqrt{\zeta_l'}$ for almost all $j$) or for most $j$ in no $T_l$ is $A_{ij}^{(2)}$ above the threshold, whence, $\mu_{ij} \leq \varepsilon_l\sqrt{\zeta_l'}$ for almost all $j$. Both these extreme cases would make us loose all the information about $T_l$ due to thresholding; this scenario and milder versions of it have to be proven not to occur. We do this by considering how thresholds handle catchwords. Indeed we will show that for a catchword $i \in S_l$, a $j \in T_l$ has $A_{ij}^{(2)}$ above the threshold and a $j \notin T_l$ has $A_{ij}^{(2)}$ below the threshold. Both statements will only hold with high probability, of course and using this, we prove that $\mu_{\cdot,j}$ and $\mu_{\cdot,j'}$ are not too close for $j, j'$ in different $T_l$ 's. For this, we need the following Lemmas.

**Lemma 3.5** For $i \in S_l$, and $l' \neq l$, we have with $\eta_i = \lfloor M_{il}(\alpha + \beta + \rho)m/2 \rfloor$,

$$Prob(A_{ij} \leq \eta_i/m \mid j \in T_l) \leq \varepsilon w_0/20, \quad Prob(A_{ij} \geq \eta_i/m \mid j \in T_{l'}) \leq \varepsilon w_0/20.$$

**Lemma 3.6** With probability at least $1 - 8mdk\exp(-\varepsilon^2 w_0 s/8)$, we have

$$for \ j \in T_l, j' \notin T_l, \ |\mu_{\cdot,j} - \mu_{\cdot,j'}|^2 \geq \frac{2}{9}\alpha p_0 m.$$

### 3.1.2 Proximity

Next, we wish to show that clustering as in TSVD identifies the dominant topics correctly for most documents, i.e., that $R_l \approx T_l$ for all $l$. For this, we will use a theorem from [3] [see also [4]] which in this context says:

**Theorem 3.7** *If all but a $f$ fraction of the the $B_{\cdot,j}$ satisfy the "proximity condition", then TSVD identifies the dominant topic in all but $c_1 f$ fraction of the documents correctly after polynomial number of iterations.*

To describe the proximity condition, first let $\sigma$ be the maximum over all directions $v$ of the square root of the mean-squared distance of $B_{\cdot,j}$ to $\mu_{\cdot,j}$, i.e.,

$$\sigma^2 = \text{Max}_{\|v\|=1} \frac{1}{s} |v^T (\mathbf{B} - \mu)|^2 = \frac{1}{s} \|\mathbf{B} - \mu\|^2.$$

The parameter $\sigma$ should remind the reader of standard deviation, which is indeed what this is, since $E(\mathbf{B}|T_1, T_2, \ldots, T_l) = \mu$. Our random variables $B_{\cdot,j}$ being $d-$ dimensional vectors, we take the maximum standard deviation in any direction.

**Definition:** $\mathbf{B}$ is said to satisfy the proximity condition with respect to $\mu$, if for each $l$ and each $j \in T_l$ and and each $l' \neq l$ and $j' \in T_{l'}$, the projection of $B_{\cdot,j}$ onto the line joining $\mu_{\cdot,j}$ and $\mu_{\cdot,j'}$ is closer to $\mu_{\cdot,j}$ by at least

$$\Delta = \frac{c_0 k}{\sqrt{w_0}} \sigma,$$

than it is to $\mu_{\cdot,j'}$. [Here, $c_0$ is a constant.]

To prove proximity, we need to bound $\sigma$. This will be the task of the subsection 4.1 which relies heavily on Random Matrix Theory.

## 4 Proofs of Correctness

We start by recalling the Höffding-Chernoff (H-C) inequality in the form we use it.

**Lemma 4.1 Höffding-Chernoff** *If $X$ is the average of $r$ independent random variables with values in $[0, 1]$ and $E(X) = \mu$, then, for an $t > 0$,*

$$Prob(X \geq \mu + t) \leq \exp\left(-\frac{t^2 r}{2(\mu + t)}\right) \; ; \; Prob(X \leq \mu - t) \leq \exp\left(-\frac{t^2 r}{2\mu}\right).$$

**Proof:** (of Lemma 3.1) Abbreviate $p_i(\cdot, l)$ by $f(\cdot)$. We claim that either (i) $f(\zeta) \geq f(\zeta - 1)\forall 1 \leq \zeta \leq \zeta_0$ or (ii) $f(\zeta + 1) \leq f(\zeta)\forall m - 1 \geq \zeta \geq \zeta_0$. To see this, note that if both (i) and (ii) fail, we have $\zeta_1 \leq \zeta_0$ and $\zeta_2 \geq \zeta_0$ with $f(\zeta_1) - f(\zeta_1 - 1) < 0 < f(\zeta_2 + 1) - f(\zeta_2)$. But then there has to be a local minimum of $f$ between $\zeta_1$ and $\zeta_2$. If (i) holds, clearly, $f(\zeta_0) \geq f(\zeta)\forall \zeta < \zeta_0$ and so the lemma follows. So, also if (ii) holds.

**Proof:** (of Lemma 3.2) Note that $q_i(\zeta, l) = \frac{1}{s} |\{j \in T_l : A_{ij} = \zeta/m\}| = \frac{1}{s} \sum_{j=1}^{s} X_j$, where, $X_j$ is the indicator variable of $A_{ij} = \zeta/m \wedge j \in T_l$. $\frac{1}{s} \sum_j E(X_j) = p_i(\zeta, l)$ and we apply H-C with $t = \frac{1}{2}\varepsilon\sqrt{w_0}\sqrt{p_i(\zeta, l)} + \frac{1}{2}\varepsilon^2 w_0$ and $\mu = p_i(\zeta, l)$. We have $\frac{t^2}{\mu + t} \geq \varepsilon^2 w_0/4$, as is easily seen by calculating the roots of the quadratic $t^2 - \frac{1}{4}t\varepsilon^2 w_0 - \frac{1}{4}\varepsilon^2 w_0\mu = 0$. Thus we get the claimed for $T_l$. Note that the same proof applies for $T_l^{(1)}$ as well as $T_l^{(2)}$.

To prove the second assertion, let $a = q_i(\zeta, l)$ and $b = \sqrt{p_i(\zeta, l)}$, then, $b$ satisfies the quadratic inequalities:

$$b^2 - \frac{1}{2}\varepsilon\sqrt{w_0}b - (a + \frac{1}{2}\varepsilon^2 w_0) \leq 0 \; ; \; b^2 + \frac{1}{2}\varepsilon\sqrt{w_0}b - (a - \frac{1}{2}\varepsilon^2 w_0) \geq 0.$$

By bounding the roots of these quadratics, it is easy to see the second assertion after some calculation.

**Proof:** (of Lemma 3.3) Note that $\zeta_i$ is a random variable which depends only on $A^{(1)}$. So, for $j \in T_l^{(2)}$, $A_{ij}$ are independent of $\zeta_i$. Now, if

$$\text{Prob}(A_{ij} \leq \frac{\zeta_i}{m} \; ; j \in T_l^{(2)}) > 4m\varepsilon w_0 \text{ and Prob}(A_{ij} > \frac{\zeta_i}{m} \; ; j \in T_l^{(2)}) > 4m\varepsilon w_0,$$

by Lemma (3.1), we have

$$\text{Prob}(A_{ij} = \frac{\zeta_i}{m}; j \in T_l^{(2)}) > 4\varepsilon w_0.$$

Since $\text{Prob}(A_{ij} = \zeta/m; j \in T_l^{(1)}) = \text{Prob}(A_{ij} = \zeta/m; j \in T_l^{(2)})$ for all $\zeta$, we also have

$$\text{Prob}(A_{ij} = \frac{\zeta_i}{m}; j \in T_l^{(1)}) = p_i(\zeta_i, l) > 4\varepsilon w_0. \tag{11}$$

Pay a failure probability of $2\exp(-\varepsilon^2 s w_0/8)$ and assume the conclusion of Lemma (3.2) and we have:

$$\frac{1}{s}\left|\{j \in T_l^{(1)} : A_{ij} = \frac{\zeta_i}{m}\}\right| = q_i(\zeta_i, l) \geq p_i(\zeta_i, l) - \frac{\varepsilon}{2}\sqrt{w_0 p_i(\zeta_i, l)} - \frac{\varepsilon^2}{2}w_0.$$

Now, it is easy to see that $p_i(\zeta, l) - \frac{\varepsilon}{2}\sqrt{w_0 p_i(\zeta, l)}$ increases as $p_i(\zeta, l)$ increases subject to (11). So,

$$p_i(\zeta, l) - \frac{\varepsilon}{2}\sqrt{w_0 p_i(\zeta, l)} - \frac{\varepsilon^2}{2}w_0 > (4\varepsilon - \varepsilon^{3/2} - \frac{1}{2}\varepsilon^2)w_0 \geq 3\varepsilon w_0,$$

contradicting the definition of $\zeta_i$ in the algorithm. This completes the proof of the Lemma.

**Proof:** (of Lemma 3.4): After paying a failure probability of $4kd\exp(-\varepsilon^2 s w_0/8)$, assume no threshold splits any $T_l$. [The factors of $k$ and $d$ come in because we are taking the union bound over all words and all topics.] Then,

$$\text{Prob}(A_{ij}^{(2)} \leq \frac{\zeta_i}{m} \mid j \in T_l^{(2)}) = \sum_{\zeta=0}^{\zeta_i} p_i(\zeta, l)/\text{Prob}(j \in T_l) \leq 4m\varepsilon\frac{w_0}{w_l}$$

$$\text{or Prob}(A_{ij}^{(2)} > \frac{\zeta_i}{m} \mid j \in T_l^{(2)}) = \sum_{\zeta=\zeta_i+1}^{m} p_i(\zeta, l)/\text{Prob}(j \in T_l) \leq 4m\varepsilon\frac{w_0}{w_l}.$$

Wlg, assume that $\text{Prob}(A_{ij} \leq \zeta_i/m \mid j \in T_l) \leq \varepsilon_l$. Then, with probability, at least $1 - \varepsilon_l$, $A_{ij}^{(2)} > \zeta_i/m$. Now, either $\zeta_i < 8\ln(20/\varepsilon w_0)$ and all $B_{ij}, j \in T_l$ are zero and then $\mu_{ij} = 0$, or $\zeta_i \geq 8\ln(20/\varepsilon w_0)$, whence, $E(B_{ij}|j \in T_l) \in [(1-\varepsilon_l)\sqrt{\zeta_i'}, \sqrt{\zeta_i'}]$. So, $\mu_{ij} \geq (1-\varepsilon_l)\sqrt{\zeta_i'}$ and $\text{Prob}(B_{ij} = 0) \leq \varepsilon_l$. So,

$$\text{Var}(B_{ij}^2|j \in T_l) \leq (\sqrt{\zeta_i'}-(1-\varepsilon_l)\sqrt{\zeta_i'})^2\text{Prob}(B_{ij} = \sqrt{\zeta_i'}|j \in T_l)+(\sqrt{\zeta_i'}-0)^2\text{Prob}(B_{ij} = 0|j \in T_l) \leq 2\varepsilon_l\zeta_i'.$$

This proves the lemma in this case. The other case is symmetric.

**Proof:** (of Lemma 3.5) Recall that $P_{ij} = \sum_l M_{il}W_{lj}$ is the probability of word $i$ in document $j$ conditioned on $\mathbf{W}$. Fix an $i \in S_l$. From the dominant topic assumption,

$$\forall j \in T_l, P_{ij} = \sum_{l_1} M_{il_1}W_{l_1,j} \geq M_{il}W_{lj} \geq M_{il}\alpha. \tag{12}$$

The $P_{ij}$ are themselves random variables. Note that (12) holds with probability 1. From Catchword assumption and (1), we get that

$$M_{il}\alpha - (\eta_i/m) \geq M_{il}\alpha - M_{il}((\alpha + \beta + \rho)/2) \geq M_{il}\alpha\delta/2.$$

Now, we will apply H-C with $\mu - t = \eta_i/m$ and $\mu \geq M_{il}\alpha$ for the $m$ independent words in a document. By Calculus, the probability bound from H-C of $\exp(-t^2 w_l s/2\mu) = \exp(-(\mu - (\eta_i/m))/2\mu)$ is highest subject to the constraints $\mu \geq M_{il}\alpha; \eta_i \leq mM_{il}(\alpha + \beta + \rho)/2$, when $\mu = M_{il}\alpha$ and $\eta_i = mM_{il}(\alpha + \beta + \rho)/2$, whence, we get

$$\text{Prob}(A_{ij} \leq \eta_i/m \mid j \in T_l) \leq \exp(-M_{il}\alpha\delta^2 m/8) \leq \varepsilon w_0/20,$$

using (5). Now, we prove the second assertion of the Lemma.

$$\forall j \in T_{l'}, l' \neq l, \sum_{l_1} M_{il_1} W_{l_1,j} = M_{il} W_{lj} + \sum_{l_1 \neq l} M_{il_1} W_{l_1,j}$$

$$\leq M_{il} W_{lj} + \left(\mathrm{Max}_{l_1 \neq l} M_{il_1}\right)(1 - W_{lj})$$

$$\leq M_{il}(\beta + \rho). \tag{13}$$

$$\frac{\eta_i}{m} - M_{il}(\beta + \rho) \geq \frac{M_{il}(\alpha + \beta + \rho)}{2} - M_{il}(\beta + \rho) - \frac{1}{m} \geq \frac{3M_{il}\alpha\delta}{8},$$

using (5) and (1). Applying the first inequality of Lemma (4.1) with $\mu + t = \eta_i/m$ and $\mu \leq M_{il}(\beta + \rho)$ and again using (5),

$$\mathrm{Prob}(A_{ij} \geq \eta_i/m \mid j \in T_{l'}) \leq \exp\left(-9M_{il}\alpha\delta^2 m/64\right) \leq \varepsilon w_0/20.$$

**Lemma 4.2** *For $i \in S_l$, $Prob(\zeta_i < \eta_i) \leq 3mke^{-\varepsilon^2 s w_0/8}$, with $\eta_i$ as defined in Lemma 3.5.*

**Proof:** Fix attention on $i \in S_l$. After paying the failure probability of $3mke^{-\varepsilon^2 s w_0/8}$, assume the conclusions of Lemma (3.2) hold for all $\zeta, l$. It suffices to show that

$$\left|\{j : A_{ij}^{(1)} > \eta_i/m\}\right| \geq \frac{w_0 s}{2} \;,\; \left|\{j : A_{ij}^{(1)} = \frac{\eta_i}{m}\}\right| < 3w_0 \varepsilon s,$$

since, $\eta_i$ is an integer and $\zeta_i$ is the largest integer satisfying the inequalities. The first statement follows from first assertion of Lemma 3.5. The second statement is slightly more complicated. Using both the first and second assertions of Lemma 3.5, we get that for all $l'$ (including $l' = l$), we have

$$\mathrm{Prob}(A_{ij} = \eta_i/m | j \in T_{l'}^{(1)}) \leq \varepsilon w_0/20.$$

$$\left|\{j \in T_{l'}^{(1)} : A_{ij} = \eta_i/m\}\right| \leq \varepsilon w_0 w_{l'} s/20 + \frac{\varepsilon}{2}\sqrt{w_0/w_{l'}}\sqrt{\varepsilon w_0/20} w_{l'} s + \frac{\varepsilon^2 w_0 w_{l'}}{2}$$

$$\leq \frac{\varepsilon w_0 s}{8}\left(w_{l'} + \sqrt{\varepsilon w_{l'}}\right) + \frac{\varepsilon^2 w_0 s}{2}.$$

Now, adding over all $l'$ and using $\sum_{l'} \sqrt{w_{l'}} \leq \sqrt{k}\sqrt{\sum_{l'} w_{l'}} = \sqrt{k}$, we get

$$\left|\{j : A_{ij}^{(1)} = \eta_i/m\}\right| \leq \varepsilon w_o s,$$

since $\varepsilon \leq 1/k$.

**Lemma 4.3** *Define $I_l = \{i \in S_l : \zeta_i \geq \eta_i\}$. With probability at least $1 - 8mdk\exp(-\varepsilon^2 w_0 s/8)$, we have for all $l$,*

$$\sum_{i \in I_l} \zeta_i' \geq m\alpha p_0/2.$$

**Proof:** After paying the failure probability, we assume the conclusion of Lemma 3.2 holds for all $i, \zeta, l$. Now, by Lemma 4.2, we have (with $\mathbf{1}$ denoting the indicator function)

$$E\left(\sum_{i \in S_l} M_{il}\mathbf{1}(\zeta_i < \eta_i)\right) \leq 3mk\exp(-\varepsilon^2 s w_0/8)\sum_{i \in S_l} M_{il},$$

which using Markov inequality implies that with probability at least $1 - 6mk\exp(-\varepsilon^2 s w_0/8)$,

$$\sum_{i \in I_l} M_{il} \geq \frac{1}{2}\sum_{i \in S_l} M_{il} \geq p_0/2, \tag{14}$$

using (4). Note that by (5), no catchword has $\zeta_i'$ set to zero. So,

$$\sum_{i \in I_l} \zeta_i' = \sum_{i \in I_l} \zeta_i \geq \sum_{i \in I_l} \eta_i \geq \sum_{I_l} m M_{il}\alpha/2 \geq \alpha p_0 m/2.$$

**Proof:** (of Lemma 3.6) For this proof, $i$ will denote an element of $I_l$. By Lemma 3.5,

$$\forall i \in I_l, l' \neq l, \text{Prob}(A_{ij} > \frac{\zeta_i}{m}|j \in T_{l'}^{(1)}) \leq \text{Prob}(A_{ij} > \eta_i/m|j \in T_{l'}^{(1)}) \leq \frac{\varepsilon w_0}{20}. \tag{15}$$

This implies by Lemma 3.2, for $l' \neq l$,

$$\left|\{j \in T_{l'}^{(1)} : A_{ij} > \frac{\zeta_i}{m}\}\right| \leq w_{l'}s\left(\frac{\varepsilon w_0}{20} + \varepsilon\sqrt{w_0/w_{l'}}\sqrt{\varepsilon w_0}/4\right) + w_0\varepsilon^2 s/2. \tag{16}$$

Summing over all $l' \neq l$, we get (using $\sum_{l'}\sqrt{w_{l'}} \leq \sqrt{\sum w_{l'}}\sqrt{k} \leq 1/\sqrt{\varepsilon}$ by (9))

$$\sum_{l'\neq l}\left|\{j \in T_{l'}^{(1)} : A_{ij} > \frac{\zeta_i}{m}\}\right| \leq \varepsilon w_0 s.$$

Now the definition of $\zeta_i$ in the algorithm implies that:

$$\sum_{\zeta > \zeta_i} q_i(\zeta, l) = \left|\{j \in T_l^{(1)} : A_{ij} > \frac{\zeta_i}{m}\}\right| \geq \left(\frac{w_0}{2} - \varepsilon w_0\right) s \geq w_0 s/4.$$

So, by Lemma 3.2,

$$\text{Prob}(j \in T_l; A_{ij} > \zeta_i/m) = \sum_{\zeta > \zeta_i} p_i(\zeta, l) \geq \frac{1}{2}\sum_{\zeta > \zeta_i} q_i(\zeta, l) - \varepsilon^2 w_0 m$$

$$\geq \frac{w_0}{8} - \varepsilon^2 w_0 m \geq w_0/9,$$

using (9). Next let $\tilde{p} = \text{Prob}(A_{ij} = \zeta_i/m; j \in T_l)$. Since $|\{j \in T_l^{(1)} : A_{ij} = \zeta_i/m\}| \leq 3\varepsilon w_0 s$, by the definition of $\zeta_i$ in the algorithm, we get by a similar argument

$$\tilde{p} \leq 2q_i(\zeta_i, l) + 2\varepsilon^2 w_0 \leq 7\varepsilon w_0. \tag{17}$$

Now, by Lemma 3.1, we have

$$\tilde{p} \geq \text{Min}\left(\frac{w_0}{9m}, \frac{1}{m}\text{Prob}(A_{ij} \leq \zeta_i/m; j \in T_l^{(2)})\right).$$

By (9), $7\varepsilon w_0 < w_0/9m$ and so we get:

$$\text{Prob}(A_{ij} \leq \zeta_i/m; j \in T_l^{(2)}) \leq 7\varepsilon m w_0.$$

Noting that by (5), no catchword has $\zeta_i'$ set to zero, $\text{Prob}(B_{ij} = 0|j \in T_l^{(2)}) \leq 7\varepsilon m w_0/w_l \leq 1/6$, by (9). This implies

$$\mu_{ij} \geq \frac{5}{6}\sqrt{\zeta_i'}.$$

Now, by (15), we have for $j' \notin T_l$,

$$\mu_{ij'} \leq \sqrt{\zeta_i'}/6.$$

So, we have

$$|\mu_{\cdot,j} - \mu_{\cdot,j'}|^2 \geq \sum_{i \in I_l}(\mu_{ij} - \mu_{ij'})^2 \geq (4/9)\sum_{i \in I_l}\zeta_i'.$$

Now Lemma (4.3) implies the current Lemma. $\blacksquare$

## 4.1 Bounding the Spectral norm

**Theorem 4.4** *Fix an $l$. For $j \in T_l$, let $R_{\cdot,j} = B_{\cdot,j} - \mu_{\cdot,j}$. [The $R_{\cdot,j}, j \in T_l$ are vector-valued random variables which are independent, even conditioned on the partition $T_1, T_2, \ldots, T_k$.] With probability at least $1 - 10mdk\exp(-\varepsilon^2 w_0 s/8)$, we have $||R||^2 \leq ckw_0\varepsilon sm^2$. Thus,*

$$||\mathbf{B} - \mu||^2 \leq c\varepsilon w_0 sm^2 k^2.$$

We will apply Random Matrix Theory, in particular the following theorem, to prove Theorem 4.4.

**Theorem 4.5** *[7, Theorem 5.44] Suppose $R$ is a $d \times r$ matrix with columns $R_{.,j}$ which are independent identical vector-valued random variables. Let $U = E(R_{.,j} R_{.,j}^T)$ be the inertial matrix of $R_{.,j}$. Suppose $|R_{.,j}| \le \nu$ always. Then, for any $t > 0$, with probability at least $1 - de^{-ct^2}$, we have[2]*

$$||R|| \le ||U||^{1/2}\sqrt{r} + t\nu.$$

We need the following Lemma first.

**Lemma 4.6** *With probability at least $1 - \exp(-s\varepsilon w_0/3)$, we have*

$$\zeta_0 \le 4m\lambda \; ; \; \sum_i \zeta_i' \le 4km \tag{18}$$

**Proof:** The probability of word $i$ in document $j$, is given by: $P_{ij} = \sum_l M_{il} W_{lj} \le \lambda_i$ (where, $\lambda_i = \max_l M_{il}$). If $\lambda_i < \frac{1}{m}\ln(20/\varepsilon w_0)$, then, $\text{Prob}(A_{ij} > (8/m)\ln(20/\varepsilon w_0)) \le \varepsilon w_0$ by H-C (since $A_{ij}$ is the average of $m$ i.i.d. trials). Let $X_j$ be the indicator function of $A_{ij} > (8/m)\ln(20/\varepsilon w_0)$. $X_j$ are independent and so using H-C, we see that with probability at least $1 - \exp(-\varepsilon w_0 s/3)$, less than $w_0 s/2$ of the $A_{ij}$ are greater $(8/m)\ln(20/\varepsilon w_0)$, whence, $\zeta_i' = 0$. So we have (using the union bound over all words):

$$\text{Prob}\left(\sum_{i:\lambda_i < (1/m)\ln(20/\varepsilon w_0)} \zeta_i' > 0\right) \le d\exp(-\varepsilon w_0 s/3).$$

If $\lambda_i \ge (1/m)\ln(20/\varepsilon w_0)$, then

$$\text{Prob}(A_{ij} > 4\lambda_i) \le e^{-\lambda_i m} \le \varepsilon w_0/2,$$

which implies by the same $X_j$ kind of argument that with probability at least $1 - \exp(-\varepsilon w_0 s/4)$, for a fixed $i$, $\zeta_i \le 4\lambda_i m$. Using the union bound over all words and adding all $i$, we get that with probability at least $1 - 2d\exp(-\varepsilon w_0 s/4)$,

$$\sum_i \zeta_i' \le 4m \sum_i \lambda_i \le 4m \sum_{i,l} M_{il} \le 4km.$$

Now we prove the bound on $\zeta_0$. For each fixed $i, j$, we have $\text{Prob}(A_{ij} \ge 4\lambda) \le e^{-m\lambda} \le \varepsilon w_0$. Now, let $Y_j$ be the indicator variable of $A_{ij} \ge 4\lambda$. The $Y_j, j = 1, 2, \ldots, s$ are independent (for each fixed $i$). So, $\text{Prob}(\zeta_i \ge 4m\lambda) \le \text{Prob}(\sum_j Y_j \ge w_0 s/2) \le e^{-\varepsilon w_o s/3}$. Using an union bound over all words, we get that $\text{Prob}(\zeta_0 > 4m\lambda) \le de^{-\varepsilon w_0/3}$ by H-C.

**Proof:** (of Theorem 4.4) First,

$$||U|| = \text{Max}_{|v|=1} E(v^T R_{.,j})^2 \le E(|R_{.,j}|^2) \le 2\varepsilon_l \sum_i \zeta_i' \le 8\varepsilon_l km,$$

by Lemma (4.6) and Lemma (3.4). We can also take $\nu = 2\sqrt{km}$ in Theorem 4.5 and with $t = \sqrt{\varepsilon m w_0 s}$, the first statement of the current theorem follows (noting $r = w_l s$). The second statement follows by just paying a factor of $k$ for the $k$ topics.

## 4.2 Proving Proximity

From Theorem (4.4), the $\sigma$ in definition 3.1.2 is $\sqrt{c\varepsilon w_0 m^2 k^2}$. So, the $\Delta$ in definition 3.1.2 is $cc_0\sqrt{\varepsilon}k^2 m$. So it suffices to prove:

**Lemma 4.7** *For $j \in T_l$ and $j' \in T_{l'}, l' \ne l$, let $\hat{B}_{.,j}$ be the projection of $B_{.,j}$ onto the line joining $\mu_{.,j}$ and $\mu_{.,j'}$. The probability that $|\hat{B}_{.,j} - \mu_{.,j'}| \le |\hat{B}_{.,j} - \mu_{.,j}| + cc_0 k^2\sqrt{\varepsilon}m$ is at most $c\varepsilon m w_0 \sqrt{k}/\sqrt{\alpha p_0}$. Hence, with probability at least $1 - cmdk\exp(-cw_0\varepsilon^2 s)$, the number of $j$ for which $B_{.,j}$ does not satisfy the proximity condition is at most $c\varepsilon_0 w_0 \delta s/10c_1$.*

**Proof:** After paying the failure probability of $cmdk \exp(-w_0 s\varepsilon^2/8)$, of Lemmas (4.6) and (3.6), assume that $\zeta_0 \leq 4m\lambda$, $|\mu_{\cdot,j} - \mu_{\cdot,j'}|^2 \geq 2\alpha m p_0/9$ and $\sum_i \zeta_i' \leq 4km$.

Let $X = (B_{\cdot,j} - \mu_{\cdot,j}) \cdot (\mu_{\cdot,j'} - \mu_{\cdot,j})$. $X$ is a random variable, whose expectation is 0 conditioned on $j \in T_l^{(2)}$.

Since $\mathrm{Prob}(B_{ij} = \sqrt{\zeta_i'}|j \in T_l) = \mu_{ij}/\sqrt{\zeta_i'}$, we have:

$$E|X| \leq E \sum_i |B_{ij} - \mu_{ij}| \, |\mu_{ij'} - \mu_{ij}|$$

$$= \sum_i \left[ (\sqrt{\zeta_i'} - \mu_{ij}) \frac{\mu_{ij}}{\sqrt{\zeta_i'}} + (1 - \frac{\mu_{ij}}{\sqrt{\zeta_i'}})\mu_{ij} \right] |\mu_{ij} - \mu_{ij'}|$$

$$\leq 2\varepsilon_l \sum_i \sqrt{\zeta_i'}|\mu_{ij} - \mu_{ij'}| \quad \text{by Lemma 3.4}$$

$$\leq 2\varepsilon_l \left( \sum_i \zeta_i' \right)^{1/2} |\mu_{\cdot,j} - \mu_{\cdot,j'}| \leq 4\varepsilon_l \sqrt{km}|\mu_{\cdot,j} - \mu_{\cdot,j'}|.$$

Now apply Markov inequality to get

$$\mathrm{Prob}(|X| \geq \frac{1}{8}|\mu_{\cdot,j} - \mu_{\cdot,j'}|^2) \leq 32\varepsilon_l \sqrt{km}/|\mu_{\cdot,j} - \mu_{\cdot,j'}| \leq 80\varepsilon_l \sqrt{k/\alpha p_0}.$$

If $|X| \leq |\mu_{\cdot,j} - \mu_{\cdot,j'}|^2/8$, then, $|\hat{B}_{\cdot,j} - \mu_{\cdot,j'}| \geq |\hat{B}_{\cdot,j} - \mu_{\cdot,j}| + 3|\mu_{\cdot,j} - \mu_{\cdot,j'}|/4 \geq |\hat{B}_{\cdot,j} - \mu_{\cdot,j}| + cc_0 k^2 \sqrt{\varepsilon}m$, by (9). This proves the first assertion of the Lemma.

The second statement of the Lemma follows by applying H-C to the random variable $\sum_j Z_j/s$, where, $Z_j$ is the indicator random variable of $B_{\cdot,j}$ not satisfying the proximity condition (and using (9).)

The last Lemma implies that the algorithm TSVD correctly identifies the dominant topic in all but at most $\varepsilon_0 w_0/10$ fraction of the documents by Theorem (3.7).

**Lemma 4.8** *With probability at least* $1 - \exp(-w_0 s\varepsilon^2/8)$, *TSVD correctly identifies the dominant topic in all but at most* $\varepsilon_0 w_0 \delta/10$ *fraction of documents in each* $T_l$.

### 4.3  Identifying Catchwords

Recall the definition of $J_l$ from Step 5a of the algorithm. The two lemmas below are roughly converses of each other which prove roughly that $J_l$ consists of those $i$ for which $M_{il}$ is strictly higher than $M_{il'}$. Using them, Lemma 4.11 says that almost all the $\varepsilon_0 w_0 s/2$ documents found in Step 6 of the algorithm are $1 - \delta$ pure for topic $l$.

**Lemma 4.9** *Let* $\nu = \gamma(1 - 2\delta)/(1 + \delta)$. *If* $i \in J_l$, *then for all* $l' \neq l$, $M_{il} \geq \nu M_{il'}$ *and* $M_{il} \geq \frac{3}{m\delta^2} \ln(20/\varepsilon w_0)$.

**Proof:** It is easy to check that the assumptions (2) and (1) imply $\nu \geq 2$. Let $i \in J_l$. By the definition of $J_l$ in the algorithm, $g(i,l) \geq (4/m\delta^2) \ln(20/\varepsilon w_0)$. Note that $P_{ij} \leq \mathrm{Max}_{l_1} M_{il_1}$ for all $j$. So,

$$\max_{l_1} M_{il_1} \geq \frac{3}{m\delta^2} \ln(20/\varepsilon w_0). \tag{19}$$

If the Lemma is false, then, for $l'$ attaining $\mathrm{Max}_{l_1 \neq l} M_{il_1}$, we have $M_{il} < \nu M_{il'}$. Recall $R_{l'}$ defined in Step 4c of the algorithm. Let

$$\hat{T}_{l'} = R_{l'} \cap (\text{ the set of } 1 - \delta \text{ pure documents in } T_{l'}).$$

Since all but $\varepsilon_0 w_0 s/10$ documents in $T_{l'}$ belong to $R_{l'}$, we have $|\hat{T}_{l'}| \geq 0.9\varepsilon_0 w_0 s$. For $j \in \hat{T}_{l'}$, $P_{ij} \geq M_{il'}W_{l'j} \geq (1 - \delta)M_{il'}$. So, $\mathrm{Prob}(A_{ij} < M_{il'}(1 - 2\delta)) \leq \exp(-m\delta^2 M_{il'}/3) \leq \varepsilon w_0/4$ using (19). Thus the number of documents in $R_{l'}$ for which $A_{ij} \geq M_{il'}(1-2\delta)$ is at least $0.9\varepsilon_0 w_0 s-$

$3\varepsilon w_0 s \geq .6\varepsilon_0 w_0 s$. This implies that with probability at least $1 - \exp(-c\varepsilon^2 s w_0)$, $g(i, l') \geq M_{il'}(1 - 2\delta)$.

Now, for $j \in T_l$, $P_{ij} \leq \text{Max}(M_{il}, M_{il'}) \leq \nu M_{il'}$. So, $\text{Prob}(A_{ij} > M_{il'}\nu(1 + \delta)) \leq \varepsilon w_0/4$, again using (19). At most $\varepsilon_0 w_0 s/10$ documents of other $T_{l_1}$, $l_1 \neq l$ are in $R_l$ (by Lemma 4.8). So, whp, $g(i, l) \leq M_{il'}\nu(1 + \delta)$ and so we have

$$g(i, l) \leq \frac{\nu(1 + \delta)}{1 - 2\delta} g(i, l'),$$

contradicting the definition of $J_l$. So, we must have that $M_{il} \geq \nu M_{il'}$ for all $l' \neq l$. The second assertion of the Lemma now follows from (19).

**Lemma 4.10** *If $M_{il} \geq \text{Max}\left(\frac{5}{m\delta^2}\ln(20/\varepsilon w_0), \text{Max}_{l' \neq l}\frac{1}{\rho} M_{il'}\right)$, then, with probability at least $1 - \exp(-c\varepsilon^2 w_0 s)$, we have that $i \in J_l$. So, $S_l \subseteq J_l$.*

**Proof:** Let $\hat{T}_l = R_l \cap$ (set of $1 - \delta$ pure documents in $T_l$). For $j \in \hat{T}_l$, $P_{ij} \geq M_{il}(1 - \delta)$ which implies that whp, (since $|\hat{T}_l| \geq 0.9\varepsilon_0 s$, again by Lemma 4.8)

$$g(i, l) \geq M_{il}(1 - 2\delta) \tag{20}$$

On the other hand, for $j \in T_{l'}$ and for $l' \neq l$, $i : M_{il'} \leq \rho M_{il}$ (hypothesis of the Lemma), $P_{ij} \leq M_{il}W_{lj} + \rho M_{il}(1 - W_{lj}) \leq M_{il}(\beta + \rho)$. So whp,

$$g(i, l') \leq M_{il}(\beta + \rho)(1 + \delta). \tag{21}$$

From (20) and (21) and hypothesis of the Lemma, it follows that

$$g(i, l) \geq \text{Max}\left(\frac{4}{m\delta^2}\ln(1/\varepsilon w_0), \frac{(1 - 2\delta)}{(1 + \delta)(\beta + \rho)} g(i, l')\right).$$

So, $i \in J_l$ as claimed. It only remains to check that $i$ in $S_l$ satisfies the hypothesis of the Lemma which is obvious.

**Lemma 4.11** *Let $\nu_l = \sum_{i \in J_l} M_{il}$ and let $L$ be the set of $\lfloor (s\varepsilon_0 w_0/2) \rfloor$ $A_{.,j}$ 's whose average is returned in Step 6 of the TSVD Algorithm as $\hat{M}_{.,l}$. With probability at least $1 - c\exp(-c\varepsilon^2 w_0 s)$, we have:*

$$\left| \frac{1}{|L|} \sum_{j \in L}(A_{.,j} - M_{.,l}) \right|_1 \leq O(\delta). \tag{22}$$

**Proof:** The proof needs care since $J_l$ is itself a random set dependent on $A^{(2)}$. To understand the proof intuitively, if we pretend that there is no conditioning of $J_l$ on $A^{(2)}$, then, basically, our arguments in Lemma 4.9 would yield this Lemma. However, we have to work harder to avoid conditioning effects. Define

$$K_l = \{i : M_{il} \geq \nu M_{il'} \forall l' \neq l; M_{il} \geq (3/m\delta^2)\ln(20/\varepsilon w_0)\}.$$

Note that $K_l$ is not a random set; it does not depend on $A$, just on $M$ which is fixed. Lemma 4.9 proved that $J_l \subseteq K_l$. Since $\sum_i M_{il} = 1$, we have $|K_l| \leq m\delta^2/3$. The probability bounds given here will be after conditioning on $\mathbf{W}$. [In other words, we prove statements of the form $\text{Prob}(\mathcal{E}|\mathbf{W}) \leq a$ which is (the usual) shorthand for: for each possible value $w$ of the matrix $W$, $\text{Prob}(\mathcal{E} \mid \mathbf{W} = w) \leq a$.] This will be possible, since, even after fixing $W$, the $A_{.,j}$ are independent, though certainly not identically distributed now, since the $W_{.,j}$ may differ.

For $i \in K_l$, we have for all $j$, $P_{ij} = \sum_{l'} M_{il'}W_{l'j} \leq M_{il}$, since, $M_{il'} \leq M_{il}/\nu \leq M_{il}/2$ for $l' \neq l$. For any $x \leq M_{il}$,

$$\text{Prob}(|A_{ij}^{(2)} - P_{ij}| \geq \delta M_{il} \mid W, P_{ij} = x) \leq 2\exp\left(-\frac{\delta^2 M_{il}^2 m}{2(1 + \delta)x}\right) \leq 2\exp\left(-\frac{m\delta^2 M_{il}}{3}\right).$$

Noting that $m\delta^2 M_{il} \geq 3\ln(20/\varepsilon w_0)$ for $i \in K_l$, we get

$$\text{Prob}(|A_{ij}^{(2)} - P_{ij}| \geq \delta M_{il} \mid W) \leq \varepsilon w_0/20.$$

Using the union bound over all $i \in K_l$ yields (for each $j \in [s]$),

$$\text{Prob}(\exists i \in K_l : |A_{ij}^{(2)} - P_{ij}| \geq \delta M_{il} \mid W) \leq \frac{m\delta^2 \varepsilon w_0}{20} \leq \frac{\varepsilon_0 w_0 \delta^2}{20},$$

by (9). Let

$$BAD = \{j : \exists i \in K_l : |A_{ij}^{(2)} - P_{ij}| \geq \delta M_{il}\}.$$

Using the independence of $A_{\cdot,j}$, (even conditioned on $W$), apply H-C to get that for the event

$$\mathcal{E} : |BAD| \geq \frac{s\varepsilon_0 w_0 \delta}{10}$$

$$\text{Prob}(\mathcal{E} \mid W) \leq 2\exp(-c\varepsilon w_0 s). \tag{23}$$

After paying the failure probability, for the rest of the proof, assume that $\neg\mathcal{E}$ holds. Let $U_l = \{j : W_{lj} \geq 1 - \delta\}$. By the dominant topic assumption, we know that $|U_l| \geq \varepsilon_0 w_0 s$. So, $|U_l \setminus BAD| \geq 4\varepsilon_0 w_0 s/5$ and we get (using (9)):

$$\forall N_l \subseteq K_l, \left|\{j : W_{lj} \geq 1 - \delta \; ; \; \sum_{i \in N_l} A_{ij}^{(2)} \geq (1 - 2\delta)\sum_{i \in N_l} M_{il}\}\right| \geq 4\varepsilon_0 w_0 s/5. \tag{24}$$

Now consider $j : W_{lj} \leq (1 - 6\delta)$ and $i \in K_l$.

$$P_{ij} \leq M_{il}W_{lj} + \sum_{l' \neq l} M_{il'}W_{l'j} \leq M_{il}(1 - 6\delta) + \frac{M_{il}}{\nu}6\delta \leq M_{il}(1 - 3\delta),$$

since by (2) and (1), we have that $\nu \geq 2$. So, for a $j$ with $W_{lj} \leq 1 - 6\delta$ to have $\sum_{i \in J_l} A_{ij}^{(2)} \geq (1 - 2\delta)\nu_l$, $j$ must be in $BAD$. This gives us

$$\forall N_l \subseteq K_l, \left|\{j : W_{lj} \leq (1 - 6\delta) \; ; \; \sum_{i \in N_l} A_{ij}^{(2)} \geq (1 - 2\delta)\sum_{i \in N_l} M_{il}\}\right| \leq \varepsilon_0 w_0 \delta s/10. \tag{25}$$

Let $L$ be the set of $\lfloor \varepsilon_0 w_0 s/2 \rfloor$ $j$ achieving the highest $\sum_{i \in J_l} A_{ij}^{(2)}$. By the above, $L$ contains at most $\varepsilon_0 \delta s/5$ $j$'s with $W_{lj} < 1 - 6\delta$, the rest being $j$ with $W_{lj} \geq 1 - 6\delta$. So are we finished with the proof - i.e., does this prove (22)? The answer is unfortunately, no. We can show from the above that $\sum_{i \in J_l} |A_{ij} - M_{il}| \leq O(\delta)$ for most $j \in L$ and so the average of $A_{\cdot,j}, j \in L$ is close to $M_{\cdot,l}$ when we restrict only to $i \in J_l$. But, on words not in $J_l$, we have not proved that the average of $A_{ij}^{(2)}, j \in L$ is close to $M_{\cdot,l}$. We will do so presently, but first note that this is not a trivial task. For example, if say, $M_{il} = \Omega(1/d)$ for all $i \notin K_l$ (or for a fraction of them) so that $\sum_{i \notin K_l} M_{il} \in \Omega(1)$, then an individual $A_{\cdot,j}$ could have $O(m)$ of the $A_{ij}, i \notin K_l$ set to $1/m$. [One copy of each of $O(m)$ words picked to be in the document.] But then we would have $|A_{\cdot,j} - M_{\cdot,l}|_1 \in \Omega(1)$ which is too much error. We will show that since we are taking the average over $L$ and not just a single document, this will not happen. But the proof is again tricky because of conditioning: both $J_l$ and $L$ depend on the data. So, to argue that the average over $L$ behaves well, we have to prove it for each possible $L$. There are at most $\binom{s}{\lfloor(\varepsilon_0 w_0 s/2)\rfloor} \leq (2/\varepsilon_0 w_0)^{\varepsilon_0 w_0 s/2}$ possible $L$'s and we will be able to take the union bound over all of them.

**Claim 4.1** *With probability at least* $1 - cmdk\exp(-c\varepsilon^2 w_0 s)$, *we have for each* $L \subseteq [s]$ *with* $|L| = \lfloor(\varepsilon_0 w_0 s/2)\rfloor$:

$$\left|\frac{1}{|L|}\sum_{j \in L}(A_{\cdot,j} - P_{\cdot,j})\right|_1 \leq O(\delta).$$

**Proof:** Let $X = \left|\frac{1}{|L|}\sum_{j \in L}(A_{\cdot,j} - P_{\cdot,j})\right|_1$. Each $A_{\cdot,j}$ is itself the average of $m$ independent choices of words. So

$$X = \left|\frac{1}{m|L|}\sum_{j \in L}\sum_{r=1}^{m}(A_{\cdot,j}^{(r)} - P_{\cdot,j})\right|_1.$$

So, $X$ is a function of $m|L|$ independent random variables. Changing any one of these arbitrarily changes $X$ by at most $1/m|L|$.

Recall the Bounded Difference inequality [5]:

**Lemma 4.12** *Let $z_1, \ldots, z_n, z_i'$ are $(n+1)$ independent random variables each taking values in $\mathcal{Z}$ and $h$ be a measurable function from $\mathcal{Z}^n$ to $\mathbb{R}$ with constants $r_i \geq 0, i \in [n]$ such that*

$$max_{z_1,\ldots,z_n,z_i' \in \mathcal{Z}}|h(z_1,\ldots,z_n) - h(z_1,\ldots,z_i',\ldots,z_n)| \leq r_i$$

*If $E(h)$ is the expectation of $h$ then Prob $\left(|h - E(h)| \geq t\right) \leq 2\exp\left(-\frac{t^2}{\sum_{i=1}^n r_i^2}\right)$.*

Using this we get
$$\text{Prob}(|X - EX| \geq c\delta) \leq \exp(-c\delta^2 \varepsilon_0 w_0 s m).$$

The "extra" $m$ in the exponent helps kill the upper bound of $(2/\varepsilon_0 w_0 s)^{\varepsilon_0 w_0 s/2}$ on the number of $L$'s and gives us
$$|X - EX| \leq O(\delta) \forall L.$$

We still have to bound $EX$. By Jenson's inequality,

$$EX \leq \frac{1}{|L|}\sum_i \left(E\left(\left(\sum_{j \in L}(A_{ij} - P_{ij})\right)^2\right)\right)^{1/2} \leq \frac{1}{|L|}\sum_i \sqrt{\sum_{j \in L_l} P_{ij}} \leq \sqrt{d}/\sqrt{|L|},$$

where, we have used the independence of $A_{.,j}$ and the fact that $E(A_{ij} - P_{ij})^2 = \text{Var}(A_{ij})$. This proves the claim.

We now bound $\left|\frac{1}{|L|}\sum_{j \in L}(P_{.,j} - M_{.,l})\right|_1$. Note that by (24) and (25), all but at most $\varepsilon_0 w_0 \delta s/10$ of the $j$'s in $L$ have $W_{lj} \geq 1 - 6\delta$, whence, we get $|P_{.,j} - M_{,l}|_1 \leq 6\delta$ for these $j$. For the $j$ with $W_{lj} < 1 - 6\delta$, we just use $|P_{.,j} - M_{,l}|_1 \leq 2$. So

$$\left|\frac{1}{|L|}\sum_{j \in L}(P_{.,j} - M_{.,l})\right|_1 \leq 6\delta + \frac{0.2\varepsilon_0 w_0 \delta s}{10|L_l|} \in O(\delta).$$

This finishes the proof of (22).

# 5 Additional Empirical Results

## 5.1 No-Local-Min Assumption

To check the no local-min assumption we consider the quantity $q_i(\zeta, l)$, in (7). Recall that $\mathbb{E}[q_i(\zeta, l)] = p_i(\zeta, l)$, we will analyze the behaviour of $q_i(\zeta, l)$ as a function of $\zeta$ for some topics $l$ and words $i$. As defined, we need a fixed $m$ to check this assumption and so we generate semi-synthetic data with a fixed $m$ from LDA model trained on the real NYT corpus. We find catchwords and the sets $\{T_l\}$ as in the catchwords assumption above and plot $q_i(\zeta, l)$ separately for some random catchwords and non-catchwords by fixing some random $l \in [k]$. Figure 1 shows the plots. As explained in 2.1, the plots are monotonically decreasing for non-catchwords and satisfy the assumption. On the other hand, the plots for catchwords are almost unimodal and also satisfy the assumption.

## 5.2 Topic Recovery on Synthetic Data

We learn the word-topic distributions ($\hat{M}$) for the semi-synthetic corpora using TSVD and the Recover algorithms of [8]. Given these learned topic distributions and the original data-generating distributions ($M$), we align the topics of $M$ and $\hat{M}$ by bipartite matching and rearrange the columns

Figure 1: Plot of $q_i(\zeta, l)$ for some random catchwords (left) and non-catchwords (right). Each of three plots for catchword is for one topic ($l$) with two random catchwords ($i$) for each topic and each plot on right is for one non-catchword ($i$) with curves for multiple topics ($l$).

| Corpus | Documents | Recover-L2 | Recover-KL | TSVD | % Improvement |
|--------|-----------|------------|------------|------|---------------|
| **NIPS** | 40,000 | 0.342 | 0.308 | **0.115** | **62.7%** |
|  | 50,000 | 0.346 | 0.308 | **0.145** | **52.9%** |
|  | 60,000 | 0.346 | 0.311 | **0.131** | **57.9%** |
| **Pubmed** | 40,000 | 0.388 | 0.332 | **0.288** | **13.3%** |
|  | 50,000 | 0.378 | 0.326 | **0.280** | **14.1%** |
|  | 60,000 | 0.372 | 0.328 | **0.284** | **13.4%** |
| **20NG** | 40,000 | 0.126 | **0.120** | 0.124 | -3.3% |
|  | 50,000 | 0.118 | 0.114 | **0.113** | **0.9%** |
|  | 60,000 | 0.114 | 0.110 | **0.106** | **3.6%** |
| **NYT** | 40,000 | 0.214 | 0.208 | **0.195** | **6.3%** |
|  | 50,000 | 0.211 | 0.206 | **0.185** | **10.2%** |
|  | 60,000 | 0.205 | 0.200 | **0.194** | **3.0%** |

Table 1: L1 reconstruction error on various semi-synthetic datasets. Last column is percent improvement over Recover-KL (best performing Recover algorithm).

of $\hat{M}$ in accordance to the matching with $M$. Topic recovery is measured by the average of the $l_1$ error across topics (called reconstruction error [8]), $\Delta(M, \hat{M})$, defined as:

$$\Delta(M, \hat{M}) = \frac{1}{k} \sum_{l=1}^{k} \sum_{i=1}^{d} |M_{il} - \hat{M}_{il}|$$

We report reconstruction error in Table 1 for TSVD and the Recover algorithms, Recover-L2 and Recover-KL. TSVD has smaller error on most datasets than the Recover-KL algorithm. We observed performance of TSVD to be always better than Recover-L2. Best performance is observed on NIPS which has largest mean document length, indicating that larger $m$ leads to better recovery. Results on 20NG are slightly worse than Recover-KL for small sample size (though better than Recover-L2), but the difference is small. While the values in Table 1 are averaged values, Figure 2 shows that TSVD algorithm achieves much better topic recovery (27% improvement in $l_1$ error over Recover-KL) for majority of the topics (>90%) on most datasets.

### 5.2.1 Topic Recovery on Real Data

**Perplexity:** A standard quantitative measure used to compare topic models and inference algorithms is perplexity [10]. Perplexity of a set of $D$ test documents, where each document $j$ consists of $m_j$ words, denoted by $\mathbf{w}_j$, is defined as: $perp(D_{test}) = exp\left\{ -\frac{\sum_{j=1}^{D} \log p(\mathbf{w}_j)}{\sum_{j=1}^{D} m_j} \right\}$. To evaluate perplexity on real data, the held-out sets consist of 350 documents for NIPS, 10000 documents for NYT and Pubmed, and 6780 documents for 20NewsGroup. Table 2 shows the results of perplexity on the 4 datasets. TSVD gives comparable perplexity with Recover-KL, results being slightly better on NYT and 20NewsGroup which are larger datasets with moderately high mean document lengths.

Figure 2: Histogram of $l_1$ error across topics for 40,000 synthetic documents. *TSVD* (blue, solid border) gets better recovery on most topics ($> 90\%$) for most datasets (leaving small number of outliers) than *Recover-KL* (green, dashed border).

| Corpus | Perplexity | | | Topic Coherence | | |
|---|---|---|---|---|---|---|
| | R-KL | R-L2 | TSVD | R-KL | R-L2 | TSVD |
| NIPS | 754 | **749** | 835 | $-86.4 \pm 24.5$ | $-88.6 \pm 22.7$ | **$-65.2 \pm 29.4$** |
| NYT | 1579 | 1685 | **1555** | $-105.2 \pm 25.0$ | **$-102.1 \pm 28.2$** | $-107.6 \pm 25.7$ |
| Pubmed | **1188** | 1203 | 1307 | $-94.0 \pm 22.5$ | $-94.4 \pm 22.5$ | **$-84.5 \pm 28.7$** |
| 20NG | 2431 | 2565 | **2390** | $-93.7 \pm 13.6$ | **$-89.4 \pm 20.7$** | $-90.4 \pm 27.0$ |

Table 2: Perplexity and Topic Coherence

**Topic Coherence:** [9] proposed Topic Coherence as a measure of semantic quality of the learned topics by approximating user experience of topic quality on top $d_0$ words of a topic. Topic coherence is defined as $TC(d_0) = \sum_{i \le d_0} \sum_{j < i} \log \frac{D(w_i, w_j) + e}{D(w_j)}$, where $D(w)$ is the document frequency of a word $w$, $D(w_i, w_j)$ is the document frequency of $w_i$ and $w_j$ together, and $e$ is a small constant. We evaluate TC for the top 5 words of the recovered topic distributions and report the average and standard deviation across topics. TSVD gives comparable results on Topic Coherence (see Table 2).

**Topics on Real Data:** Table 3 shows the top 5 words of all 50 matched pair of topics on NYT dataset for TSVD, Recover-KL and Gibbs sampling. Most of the topics recovered by TSVD are more closer to Gibbs sampling topics. Indeed, the total average $l_1$ error with topics from Gibbs sampling for topics from TSVD is 0.034, whereas for Recover-KL it is 0.047 (on the NYT dataset).

Table 3: Top 5 words of matched topic pairs for TSVD, Recover-KL and Gibbs sampling. Catchwords and anchor words in top 5 words are highlighted for TSVD and Recover-KL

| TSVD | Recover-KL | Gibbs |
|---|---|---|
| **zzz_elian zzz_miami boy father zzz_cuba** | zzz_elian boy zzz_miami father family | zzz_elian zzz_miami boy father zzz_cuba |
| **cup minutes add tablespoon oil** | cup minutes **tablespoon** add oil | cup minutes add tablespoon oil |
| game team **yard zzz_ram** season | game team season play **zzz_ram** | team season game coach zzz_nfl |
| book find **british** sales **retailer** | book find school woman women | book find woman british school |
| | | *Continued on next page* |

Table 3: Top 5 words of matched topic pairs for TSVD, Recover-KL and Gibbs sampling. Catch-words and anchor words in top 5 words are highlighted for TSVD and Recover-KL

| TSVD | Recover-KL | Gibbs |
|---|---|---|
| **run inning hit** season game | run season game **inning** hit | run season game hit inning |
| **church zzz_god religious jewish christian** | **pope** church book jewish religious | religious church jewish jew zzz_god |
| **patient drug doctor cancer medical** | patient drug doctor percent found | patient doctor drug medical cancer |
| **music song album musical band** | black reporter zzz_new_york zzz_black show | music song album band musical |
| **computer software** system zzz_microsoft company | web www site **cookie** cookies | computer system software technology mail |
| **house dog water hair** look | room show look home house | room look water house hand |
| **zzz_china** trade zzz_united_states **nuclear** official | zzz_china **zzz_taiwan** government trade zzz_party | zzz_china zzz_united_states zzz_u_s zzz_clinton zzz_american |
| **zzz_russian war rebel troop military** | zzz_russian zzz_russia war zzz_vladimir_putin rebel | war military zzz_russian soldier troop |
| **officer police case lawyer trial** | **zzz_ray_lewis** police case officer death | police officer official case investigation |
| **car** driver **wheel** race **vehicles** | car driver truck system model | car driver truck vehicle wheel |
| **show network zzz_abc zzz_nbc viewer** | **con** zzz_mexico son federal mayor | show television network series zzz_abc |
| **com question information zzz_eastern sport** | com information question zzz_eastern sport | com information daily question zzz_eastern |
| **book** author writer com **reader** | **zzz_john_rocker** player team right braves | book word writer author wrote |
| **zzz_al_gore** zzz_bill_bradley campaign president democratic | zzz_al_gore **zzz_bill_bradley** campaign president percent | zzz_al_gore campaign zzz_bill_bradley president democratic |
| **actor** film play movie character | goal play team season game | film movie award actor zzz_oscar |
| **school student teacher district** program | school student program million children | school student teacher program children |
| **tax taxes cut** billion plan | **zzz_governor_bush** tax campaign taxes plan | tax plan billion million cut |
| **percent stock market fund investor** | million percent tax bond fund | stock market percent fund investor |
| team player season coach zzz_nfl | team season player coach **zzz_cowboy** | team player season coach league |
| family home friend room school | look gun game point shot | family home father son friend |
| **primary** zzz_mccain voter zzz_john_mccain zzz_bush | **zzz_john_mccain** zzz_george_bush campaign republican voter | zzz_john_mccain zzz_george_bush campaign zzz_bush zzz_mccain |
| **zzz_microsoft court** company case law | **zzz_microsoft** company computer system software | zzz_microsoft company window antitrust government |
| **company** million percent **shares billion** | million company stock percent **shares** | company million companies business market |
| **site web sites** com www | web site zzz_internet company com | web site zzz_internet online sites |
| **scientist human cell study researcher** | dog quick jump **altered** food | plant human food product scientist |
| **baby mom** percent home family | **mate** women bird film idea | women look com need telegram |
| | | *Continued on next page* |

Table 3: Top 5 words of matched topic pairs for TSVD, Recover-KL and Gibbs sampling. Catch-words and anchor words in top 5 words are highlighted for TSVD and Recover-KL

| TSVD | Recover-KL | Gibbs |
|---|---|---|
| **point** game **half** shot team | point game team season zzz_laker | game point team play season |
| **zzz_russia zzz_vladimir_putin** zzz_russian **zzz_boris_yeltsin zzz_moscow** | zzz_clinton government **zzz_pakistan** zzz_india zzz_united_states | government political election zzz_vladimir_putin zzz_russia |
| com **zzz_canada** www **fax** information | **chocolate** food wine flavor buy | www com hotel room tour |
| **room restaurant** building **fish painting** | zzz_kosovo police **zzz_serb** war official | building town area resident million |
| **loved** family show friend play | film show movie music book | film movie character play director |
| **prices** percent **worker** oil price | percent stock market economy prices | percent prices economy market oil |
| **million test** shares **air president** | air wind snow **shower** weather | water snow weather air scientist |
| **zzz_clinton flag official federal zzz_white_house** | **zzz_bradley** zzz_al_gore campaign zzz_gore zzz_clinton | zzz_clinton president gay mayor zzz_rudolph_giuliani |
| **files** article computer art ball | show film country right women | art artist painting museum show |
| **con** percent zzz_mexico federal official | official **zzz_iraq** government zzz_united_states oil | zzz_mexico drug government zzz_united_states mexican |
| **involving** book film case right | **test** women study student found | plane flight passenger pilot zzz_boeing |
| **zzz_internet companies** company **business customer** | company companies deal zzz_internet **zzz_time_warner** | media zzz_time_warner television newspaper cable |
| zzz_internet companies company business customer | newspaper **zzz_chronicle** zzz_examiner zzz_hearst million | million money worker company pay |
| **goal play games king** game | **zzz_tiger_wood** shot tournament tour player | zzz_tiger_wood tour tournament shot player |
| **zzz_american** zzz_united_states **zzz_nato camp** war | zzz_israel **zzz_lebanon** peace zzz_syria israeli | zzz_israel peace palestinian talk israeli |
| **team** season game player play | team game point season player | race won win fight team |
| **reporter zzz_earl_caldwell** zzz_black black look | **corp** group list oil meeting | black white zzz_black hispanic reporter |
| campaign **zzz_republican** republican **zzz_party** primary | **zzz_bush** zzz_mccain campaign republican voter | gun bill law zzz_congress legislation |
| **zzz_bush zzz_mccain campaign** primary republican | flag black **zzz_confederate** right group | flag zzz_confederate zzz_south_carolina black zzz_south |
| **zzz_john_mccain** campaign zzz_george_bush zzz_bush republican | official government case officer security | court law case lawyer right |

## Footnotes

[1] The superscript $^*$ hides a logarithmic factor in $dsk/\delta_{\text{fail}}$, where, $\delta_{\text{fail}} > 0$ is the desired upper bound on the probability of failure.

[2]$||R||$ denotes the spectral norm of $R$.