[Reviews · NeurIPS 2014]

Submitted by Assigned_Reviewer_6

The paper extends the work of Arora et al by relaxing one assumption and adding another. Topics can have multiple distinctive words rather than a single distinctive word, and each document has a single dominant topic. The authors show that an SVD on a truncated version of the document-words matrix can reconstruct topics under these assumptions to within a provable bound, assuming that the model is valid.

I like that the authors attempt to validate the assumptions. I'm surprised how many documents have single dominant topics.

"Topics" are not guaranteed to have any connection to human notions of corpus organization. That is, don't assume you should run a 20 topic model on 20 newsgroups. 20 may not be a bad number, but it's extremely unlikely to recover the original newsgroups, which are not very distinct anyway.

Since this is NIPS and not COLT, I'd prefer to see theorems and empirical results in the main paper and the proof of correctness in the supplementary material. Especially since the two pages are simply sketches of real proofs in the supplement anyway, I don't really see any reason not to include the full panel of results.

I'm concerned about scalability. One of the main advantages of other spectral methods is that they summarize the input corpus as a matrix of second or third order word statistics, so the resulting algorithms are independent of the size of the number of documents. This method keeps those documents, and thus could face a prohibitively large SVD. The comments in the conclusion are not reassuring!

On the other hand, the importance of k-means clustering of documents makes me wonder whether this work could be used to analyze a simple algorithm, like partitioning the documents and finding the characteristic words of each cluster? If single or mostly-single membership is so common, maybe thresholding is enough to make this work.
Summary: In order to make provable guarantees for NP-Hard problems like topic modeling, we need to make specific assumptions.
This work offers new assumptions on document-topic distributions and relaxed assumptions on words within topics.
The paper could be more balanced toward empirical results, and there are concerns about practical performance.

Submitted by Assigned_Reviewer_18

Overall, a very well-written paper for an interesting approach to topic recovery. The paper is focused on a method with provable recovery properties, and I think they advance the state-of-the-art in this respect.

I appreciate that the constraints are considerably weaker than competing approaches. However, the dominant topic assumption seems to me to still be a bit suspect. In particular, it is harder to satisfy in real data as the number of documents, s, increases. And in the authors' studies in 4.1 show, this is frequently violated. I think it would've been nice for the authors to include among the semi-synthetic data sets and evaluation of how the results vary as one removes/includes documents violating this constraint.

The algorithm also seems to depend critically on w0 and epsilon. In the experimental section those values were magically chosen. How sensitive are the results to these choices (especially since w0 in principle is a property of the corpus). Also the fact that the authors had to change the value of epsilon for different sections of the approach strikes me as odd. Is there some intuition for what or how those values were chosen? What is the difference in performance if only one value is chosen.

In Figure 1, it would be worth comparing to the theoretical bound since the authors can compute it on the synthetic data. How tight are the bounds in practice?

Finally, I wonder why the authors' approach does not improve upon 20 newsgroups or perplexity/coherence on the real data sets? Does this imply that the assumptions the authors make are sufficiently violated as to render many of the benefits of the theoretical analysis moot.

Nits:
Line 101: M_{i_l} should be M_{il}.
Line 135: Are the constants supposed to be 0.5 instead of 0.4? The constraints don't make much sense to me if they are 0.4.
Line 155: I wasn't entirely sure what f(\zeta) was defined as. Is the conditioning meant to imply that f is the marginal over all the j's?
Section 4.1: I assume the Gibbs sampling was using LDA but it would be worth mentioning this along with the hyperparameters.
Summary: The authors present a novel and theoretically strong approach to recovering topics from admixtures that makes relatively weak assumptions on the data as compared to related work. The main suggestion I would have are just some refinements to the quantitative evaluation.

Submitted by Assigned_Reviewer_45

This paper describes an SVD-based method for provably recovering the word-topic distributions for collections of documents taken from "dominant admixtures". This paper incrementally improves upon [5] by removing the assumption of topic "anchor words" (high probability in one topic and 0 probability in other topics) -- instead, there are weaker assumptions of topic "catchwords" (groups of words that occur strictly more frequently in one topic) and "dominant admixtures". The assumption behind dominant admixtures is that there is a topic within each document that stands out among other topics.

While I did not fully follow the proof of Theorem 3.1, I think the paper advances the line of research that solves LDA using SVD-type techniques. The clarity of the paper could improve greatly by explaining the implications and important points of the theorems, assumptions, and lemmas.

One critique I had was that this paper did seem to be incremental relative to [5] and the authors only performed empirical comparisons versus the Recover algorithm in [5]. Why not show the reconstruction errors and perplexity numbers for Gibbs sampling as well? Most topic-modelers are more familiar with the traditional approximate inference schemes such as Gibbs sampling and variational inference, and it is important to compare to these well-known algorithms as well so that the community can get a sense of the performance of the proposed algorithm.

I also did not fully understand the implications of the topic recovery evaluation. It seems the original "ground-truth" M comes from LDA Gibbs sampling on the real data. Isn't it expected a priori that the estimated M_TSVD would be closer to M than M_Recover because M_Recover is more sparse due to the Anchor word assumption? In other words, if the data-generating distribution M were modified to be highly sparse, perhaps M_Recover would be produce lower reconstruction errors. Also, the perplexity results on real data are not consistently better than Recover, so what is the evidence that this new approach has better performance?

Other comments/questions:
1) The generative process of dominant admixtures was not clearly spelled out in the Notation section.
2) Please clarify that A_ij is fractional. Line 085 says that it is a frequency.
3) Did you try the approach on a data set where most of the documents did not have a dominant topic? One way to create such a data set is to just fuse multiple documents (of completely different topics) into a super-document. How would the approach perform in this regime?
4) In Table 2, the NIPS results have lower error than the other data sets. Why not use a mean document length of 1000 for each of these semi-synthetic corpora to see if the performance can be replicated for all of these data set?
5) Cited paper [7] also removes the Anchor word assumption, but there was no comparison made between that work and the author's work.
Summary: This paper describes an SVD-based method for provably recovering the word-topic distributions for collections of documents taken from "dominant admixtures". While I did not fully follow the proof of Theorem 3.1, I think the paper advances the line of research that solves LDA using SVD-type techniques, but I have some questions about the empirical evaluation of the algorithm.
Author Feedback
Author rebuttal: We thank the referees for appreciating our contributions and respond to their points one by one.

Reviewer 45
Q1) "One critique I had was that this paper did seem to be incremental relative to [5]."
R1) Note that [5] is based on [4].
In [4] it was noted that one needs to go beyond SVD based procedures for recovering Topic matrices. In this paper we show that computing SVD on a thresholded data-matrix and subsequent k-means clustering can "provably" recover the topic matrix under weaker assumptions than[4,5]. The assumptions are different and are empirically verified on several real world corpora. The algorithm though simple requires sophisticated techniques from random matrix theory to prove correctness.
The TSVD algorithm presented here and the algorithm in [4,5] are both rigorously provable but apart from that there is nothing else in common and surely it is not an incremental improvement over [4,5].

Q2) "Isn't it expected a priori that the estimated M_TSVD would be closer to M than M_Recover because M_Recover is more sparse due to the Anchor word assumption?"
R2) M_Recover [5] does not yield sparse topic vectors even with Anchor word assumption. As defined, columns of M are topic vectors. The anchor words make some K rows of M matrix sparse.
Also note that the evaluation methodology is same as that used by [5].

Q3) "Also, the perplexity results on real data are not consistently better than Recover, so what is the evidence that this new approach has better performance?"
R3) The evidence is the L_1 error as well as the list of words for each topic. Yes, our perplexity is not consistently better, but neither is it worse. Moreover, the topic coherence results on real data are significantly better than Recover for most datasets (supplementary Table 2), as is also gauged qualitatively from the list of topic on real data (supplementary Table 4)

Q4) "Cited paper [7] also removes the Anchor word assumption, but there was no comparison made between that work and the author's work."
R4) [7] assumes however that topics are uncorrelated, whereas we do not need that assumption (see line no 57-58). Also [5], though it makes anchor word assumption, but does not need topic independence. Thus our work is closer to [5] than [7].

Reviewer 18
Q5) "However, the dominant topic assumption seems to me to still be a bit suspect ... it is harder to satisfy in real data as the number of documents, s, increases."
R5) Empirical results on real corpora (Section 4.1, Table 1) suggests that it is indeed well satisfied. The point about large s is well taken. We think that the proof should go through even if only a good fraction of documents have dominant topics, however this is still open. In the experiments(section 4.2), only a fraction satisfied the assumption, but the algorithm worked.

Q6) "I think it would've been nice for the authors to include among the semi­synthetic data sets an evaluation of how the results vary as one removes/includes documents violating this constraint."
R6) Indeed we have experimental results on this, but did not add them. We will add these results to supplementary.

Q7) "The algorithm also seems to depend critically on w0 and epsilon .... How sensitive are the results to these choices."
R7) Note that on all datasets same values of w_0 and \epsilon are used (line 320).
Following standard practice w_0 is chosen to be 1/K, K is the number of topics. Dominant topic assumption should work even better when w_0 is low as the topic weights tend to be sparse. The \epsilon parameter determines the threshold and is key to the success of TSVD. We tested for multiple values keeping this in mind and the chosen value gave best results.

Q8) "Also the fact that the authors had to change the value of epsilon for different sections of the approach strikes me as odd. Is there some intuition for what or how those values were chosen?"
R8) We believe that this is an artifact of the algorithm. The \epsilon in step 6 (line 213) depends on the number of nearly pure documents the corpus has. The epsilon in step 2a determines the threshold, which depends on how distinct the f_{i,l}'s are. Will make these points explicit in the final draft.

Q9) "Finally, I wonder why the authors' approach does not improve upon 20 newsgroups or perplexity/coherence on the real data sets?"
R9) We will probe whether the assumptions are violated. Note that our perplexity/coherence results though not better but are comparable and list of words, see supplementary, are indeed very good.

Reviewer 6
Q10) "I'm concerned about scalability. One of the main advantages of other spectral methods is that they summarize the input corpus as a matrix of second or third order word statistics ... This method keeps those documents, and thus could face a prohibitively large SVD."
R10) A point which is easy to address and we should have done it in the paper. While B is "d \times s" matrix, we only need the left singular vectors of B which are the eigen-vectors of BB^T. So we could (and should) find BB^T at the outset which is "d \times d" and only do eigen decomposition on this matrix. So except for computing BB^T (which are second order statistics and btw, [5] also does this), the time for SVD does not grow with s, only with d, that is vocabulary size (which is smaller, especially if we prune away small frequency words at the start). But indeed, your point is well taken and we should include this point in the final draft.